# Non-destructive test-based assessment of uniaxial compressive strength and elasticity modulus of intact carbonate rocks using stacking ensemble models

Davood Fereidooni[1]*, Zohre Karimi[2], Fatemeh Ghasemi[1]

**1** School of Earth Sciences, Damghan University, Damghan, Semnan, Iran, **2** School of Engineering, Damghan University, Damghan, Semnan, Iran

* d.fereidooni@du.ac.ir

**Data Availability Statement:** The source code and dataset to reproduce the results presented in this paper is available at the link: https://giHub.com/

## Abstract

The uniaxial compressive strength (UCS) and elasticity modulus (E) of intact rock are two fundamental requirements in engineering applications. These parameters can be measured either directly from the uniaxial compressive strength test or indirectly by using soft computing predictive models. In the present research, the UCS and E of intact carbonate rocks have been predicted by introducing two stacking ensemble learning models from non-destructive simple laboratory test results. For this purpose, dry unit weight, porosity, P-wave velocity, Brinell surface harnesses, UCS, and static E were measured for 70 carbonate rock samples. Then, two stacking ensemble learning models were developed for estimating the UCS and E of the rocks. The applied stacking ensemble learning method integrates the advantages of two base models in the first level, where base models are multi-layer perceptron (MLP) and random forest (RF) for predicting UCS, and support vector regressor (SVR) and extreme gradient boosting (XGBoost) for predicting E. Grid search integrating k-fold cross validation is applied to tune the parameters of both base models and meta-learner. The results demonstrate the generalization ability of the stacking ensemble method in the comparison of base models in the terms of common performance measures. The values of coefficient of determination ($R^2$) obtained from the stacking ensemble are 0.909 and 0.831 for predicting UCS and E, respectively. Similarly, the stacking ensemble yielded Root Mean Squared Error (RMSE) values of 1.967 and 0.621 for the prediction of UCS and E, respectively. Accordingly, the proposed models have superiority in the comparison of SVR and MLP as single models and RF and XGBoost as two representative ensemble models. Furthermore, sensitivity analysis is carried out to investigate the impact of input parameters.

## 1. Introduction

The uniaxial compressive strength (UCS) and elasticity modulus (E) of intact rock are two fundamental requirements for evaluating the strength, stability, and deformation behavior of rock

karimizohre/SERock. The code is written in python language and run in the jupyter notebook.

**Funding:** The author(s) received no specific funding for this work.

**Competing interests:** The authors declare that they have no known competing financial interests or personal relationships that could have appeared to influence the work reported in this paper.

**Abbreviations:** $L_s$, second level algorithm in stacking ensemble; $L_i$, first level algorithm in stacking ensemble; $M_i$, $i$th base model; $M_s$, met-model; xid, $d$th feature of $i$th data; xmaxd, maximum value of $d$th feature on all input data; xmind, minimum value of $d$th feature on all input data; zid, normalized value of xid; Adam, (adaptive moment estimation; ANFIS, adoptive neuro-fuzzy inference system; ANN, artificial neural network; D, The number of input dimensions (features); E, Elasticity modulus (GPa); HBN, Brinell hardness number (kgf/mm$^2$); k, The number of folds in k-fold cross-validation; KNN, k-nearest neighbor; L-BFGS, Limited-Memory Broyden-Fletcher-Goldfarb-Shanno; m, data count; MAE, Mean Absolute Error; MLP, Multiple Layer Perceptron; MLR, Multiple Linear Regression; MSE, Mean Squared Error; $n_e$, Effective porosity (%); R$^2$, Determination coefficients; RF, Random Forest; RMSE, Root Mean Squared Error; RSE, Relative Strength of Effect; SVR, Support Vector Regression; UCS, Uniaxial compressive strength (MPa); $v_p$, P-wave velocity (m/s); XGBoost, eXtreme Gradient Boosting; $\gamma_d$, Dry unit weight (kN/m$^3$); $d$, $d$th feature; l, the number of base models in stacking ensemble.

in engineering projects. These parameters are obtained from a common and the most important rock mechanics laboratory test namely uniaxial compressive strength test. The UCS is a main input parameter in intact rock and rock mass classifications as well as failure criteria, and the E is a parameter to identify rock stiffness and deformability of geomaterials especially rocks. Direct measurement of the UCS and E in accordance with the recommended standards such as ISRM (International Society for Rock Mechanics) and ASTM (American Society for Testing and Materials) is difficult, time-consuming, expensive, and even impossible in weak, highly fractured, inherently anisotropic, highly foliated, and stratified or laminated rocks due to preparing suitable core specimens for performing the test. For this reason, the predictive approaches are often applied for the indirect estimation of the UCS and E. Recently, various machine learning and soft computing approaches have been used to predict the two mentioned parameters based on simple laboratory tests results. In this regard, the neural networks, support vector regression, random forest, extreme gradient boosting, multiple layer perceptron, fuzzy systems, evolutionary algorithms, etc. are common predictive approaches, and predictor rock properties such as unit weight, porosity, P-wave velocity, slake durability index, and rock surface harnesses are applicable parameters [1–8]. The machine learning and soft computing approaches, unlike traditional statistical methods such as simple and multiple regressions which must be used in similar rocks, are sufficiently generic to use in various rock types because they fit all value ranges of the UCS and E to other rock properties. So, they are very suitable for the general applications as well as they reliable for all rock types.

In the last few years, some researchers applied the machine learning and soft computing-based techniques to estimate both UCS and E of various rock types [e.g. 9–23]. In this regard, Ghasemi et al. [24] applied model trees as a predicting approach and Schmidt hardness, effective porosity, dry unit weight, P-wave velocity, and slake durability index as input variables for predicting the UCS and E. They found that pruned and unpruned model trees provide suitable predictions of the parameters. Beiki et al. [18] predicted the UCS and E of carbonate rocks using genetic programming and multiple nonlinear regression models and found that the first method fitted the data more accurately than the second one, so it is the usefulness technique for estimating the UCS and E. Madhubabu et al. [25] and Aboutaleb et al. [19] have been used the MLR, ANN, and SVR for predicting both UCS and E of carbonate rocks together the R$^2$ and RMSE to examine the accuracy of the results. Their studies revealed that the ANN and SVR has a better predictive efficiency than the MLR for predicting the UCS and E from physical and index characteristics of the rocks. In another study, Rezaei and Asadizadeh [20] paid attention to the application of intelligent techniques combinations including ANFIS, genetic algorithm (GA), and particle swarm optimization (PSO) in order to predict the UCS of very strong rock types. Their studies proved that the combinations of the methods have higher capability than the regression model. Also, they found that the density and Schmidt rebound hardness had more related to the UCS than the rock porosity. Khan et al. [26] applied the MLR, ANN, RF, and KNN for predicting the UCS and static E from physical, chemical, and mechanical properties of marble rock namely density, porosity, P-wave velocity, and dynamic E under different thermal conditions. They found that the KNN and RF are reliable approaches to predict both UCS and E. Also, it was found that P-wave velocity has strong correlations with the UCS and E. Based on predictive performance, the RF model was proposed to predict the UCS and E as the best model. Shahani et al. [21, 27] in a comprehensive study measured the UCS, E, dry and wet densities, and Brazilian tensile strength of soft sedimentary rocks and predicted the UCS and E by employing the MLR models, ANN, and ANFIS from other rock parameters. Their research indicated that the approaches are suitable ways to predict the UCS and E. It is also revealed that the prediction accuracy of the ANFIS is the best among all the employed models.

Rukhaiyar and Samadhiya [28] performed a polyaxial strength model for intact sandstone based on artificial neural network (ANN) to find out the influence of each independent parameter namely uniaxial compressive strength (UCS), minor principal stress ($\sigma_3$), and intermediate principal stress ($\sigma_2$) on the strength of sandstone, i.e., the major principal stress at failure ($\sigma_1$). They found that the ANN based failure model gives the best result amongst all the considered polyaxial strength criteria, for the testing dataset. Behzadafshar et al. [29] proposed a new artificial neural network (ANN) model to approximate the elasticity modulus (E) of granite rock samples based on laboratory tests results. In this research, Rock index tests including point load, p-wave velocity and Schmidt hammer together with uniaxial compressive strength (UCS) tests were carried out to prepare a database comprised of 62 datasets for the analysis. Based on sensitivity analysis results for the developed ANN model, p-wave velocity has the most effect on E of the rock samples.

It can be easily understood from the mentioned studies that 1) statistical and soft computing approaches are suitable methods to predict engineering characteristics of rocks such as the UCS and E. 2) the machine learning and soft computing approaches are provided more accurate results than statistical methods such as simple and multiple regressions. 3) Combining several models in a proper way achieves a better result compared to only one model.

The purpose of this paper is to create an ensemble machine learning method for estimating UCS and E, based on existing research that shows how the combination of certain machine learning methods can improve their performance. However, the commonly used combinations frameworks are typically boosting and bagging. Boosting is vulnerable to overfitting and neither of these frameworks utilizes all the available data for learning base models. This paper seeks to utilize the knowledge contained within all the data by developing a stacking ensemble model to improve prediction accuracy. A little study in predicting rock brightness is done, Koopialipoor et al. [30] developed a stacking structure by employing the MLP, KNN, and RF for predicting the E from other rock properties namely porosity, Schmidt rebound hardness, P-wave velocity, and point load index. So, in the present research an attempt has been made to examine a stacking ensemble learning method for predicting the UCS and E of travertines and limestones as two major category of carbonate rocks which are one of the most abundant and common rock types in earth surface and they are encountered in many engineering projects worldwide. The contribution of this study can be listed as 1) assessing the engineering properties of 70 carbonate rock samples including 30 travertines and 40 limestones. 2) two stacking ensemble models are developed to predict E and UCS by enhancing some base models. The parameter tunning of the models is done by grid search. 3) Sensitivity analysis is performed to assess the relative importance of input variables on E and UCS. 4) extensive numerical experiments are conducted to compare the effectiveness of the proposed method with some popular learning methods. The results confirm that the proposed method outperform Support Vector Regressor (SVR), Random Forest (RF), extreme gradient boosting (XGBoost), and Multi-Layer Perceptron (MLP).

## 2. Materials and experimental studies

Providing process of necessary rock materials and their characteristics is planned in three steps namely sample selection and specimen preparation, experimental procedures and rock characteristics (laboratory investigations) as well as desk studies. Methodology flowchart of the research is presented in Fig 1. The former step includes operations for selecting suitable and applicable carbonate building stones extracted from quarries and used in many cities of Iran. Then, rock specimens with suitable shapes and dimensions were prepared for considered laboratory tests. The second step includes a comprehensive laboratory test program for evaluating

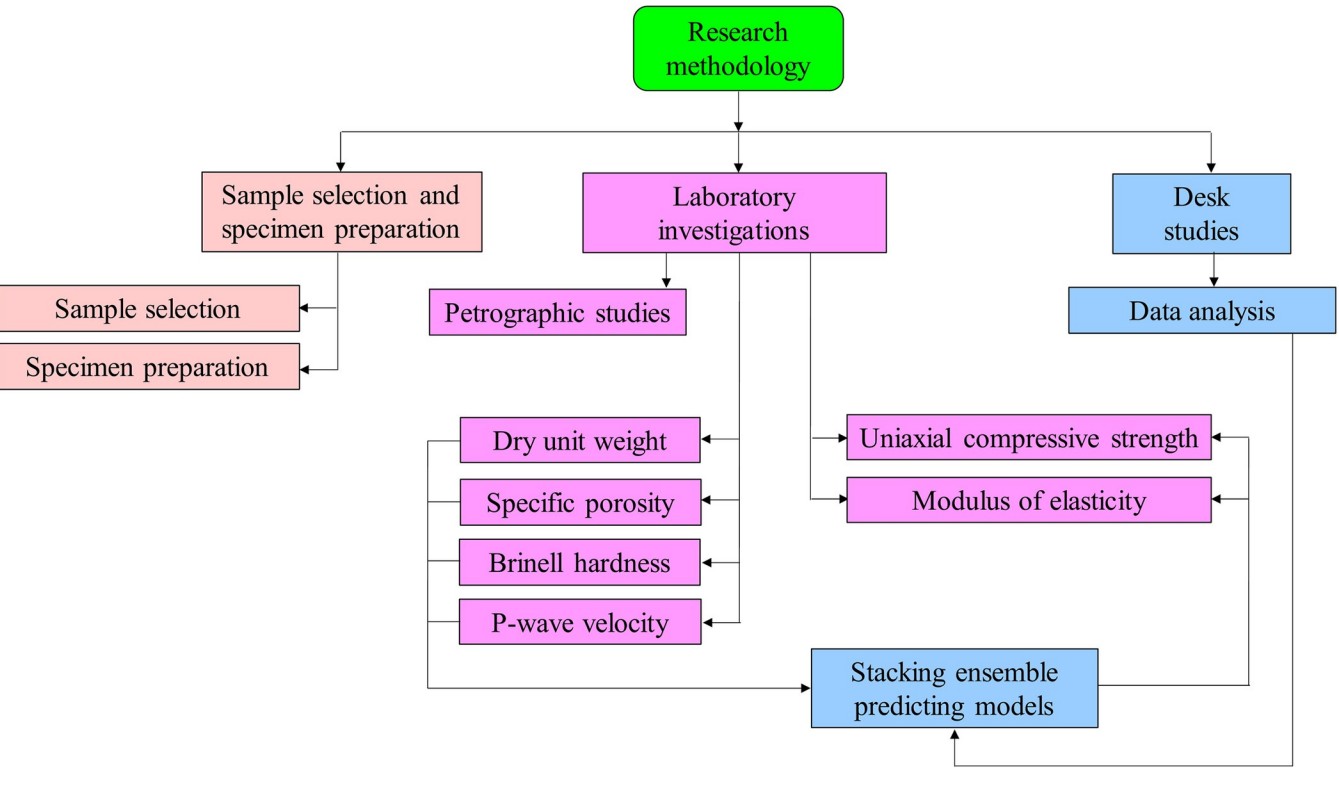

**Fig 1. Methodology flowchart of the research.**

engineering characteristics and behaviors of the selected samples. In the third step, data analysis was performed and uniaxial compressive strength and modulus of elasticity were predicted using stacking ensemble models.

## 2.1. Sample selection and specimen preparation

The number of 70 carbonate rock samples including 30 travertines and 40 limestones were taken from some quarries in different region of Iran, and moved to engineering geology laboratory. They are various in apparent properties such as color, luster, surface texture, etc. The travertines are light brown to gray in color, without any vein in surface, and with cavities filled by light brown to black materials in handy specimens. The limestones are white to cream or gray in color, with small, thin and light to dark veins, and without cavities in their surfaces. In the specimen preparation step, cube-shaped specimens with dimensions of 54×54×135mm were prepared from the selected samples in a stone cutting workshop before laboratory tests. The prepared specimens were washed with water to remove dust deposited on the stone surfaces during rock cutting and then have been dried at 105˚C in oven, and were weighed by a scale with accuracy of 0.01 g. Finally, they were tested in dry condition in accordance with ASTM [31] and ISRM [32]. Also, necessary thin sections were provided to investigate petrographic properties of the rock samples.

## 2.2. Experimental procedures and rock characteristics

The mineralogical and petrographic studies, physical properties tests (dry unit weight and effective porosity), ultrasonic wave velocity test, and Brinell hardness test were carried out for determining considered characteristics.

The microscopy studies on thin sections were done in accordance with ASTM [33] and ISRM [32] to identify mineral content, texture, and petrographic properties of the rock samples. Fig 2 shows four microscopic images of some tested rocks as representative samples in polarized light (XPL). In the travertines, the samples are consisted of micrite and sparry calcite. The sparry calcite crystals are grown in internal walls of cavities. Some of the cavities are completely filled. In the limestones, the sample are consisted of sparry calcite or micrite. In the samples that composed of micrite, the sparry calcite crystals are grown in internal walls of cavities and some of the cavities are completely filled. There are fractures and veins filled by calcite crystals and iron dioxide. Some sample have fossil.

Physical properties tests were carried out on the prepared specimens with regular shape method to determine dry unit weight ($\gamma d$) and effective porosity (ne) in accordance with ASTM [31] and ISRM [32]. Four number of experiments were used for each rock sample and averaging was done from the four obtained numbers. The samples that showed wrong results were repeated again to reduce the test error. So, a total of 280 specimens were tested in this step to determine the average values of $\gamma_d$ and $n_e$. Ultrasonic wave velocity (P-wave velocity) of the selected rock samples was determined in the laboratory using a Peroceq digital ultrasonic tester (Model: ND 180; Trade name; Pundit Lab+; Transit time range: 0.1–9999 µs; Energizing pulse: 125, 250, 350, 500 V; Frequency range: 24–500 KHz) in accordance with ASTM [34] and ISRM [32]. The Brinell hardness test is performed based on ASTM [35]. The test method is generally used to test materials with coarse structure and rough surface. In this test method, a constant load (F) is applied to a carbide ball during a predetermined time period. Then, the created impression of the ball is measured with an optical system at two perpendicular diameters. The Brinell hardness number is calculated as:

$$BHN = \frac{2F}{\pi D_b(D_b - \sqrt{(D_b^2 - D_i^2)})} \qquad (1)$$

where BHN is Brinell hardness number in kgf/mm$^2$, F is applied load in kgf (F = 1000 kgf), $D_b$ is diameter of indenter ball ($D_b$ = 10 mm), and $D_i$ is diameter of impression (mm).

Uniaxial compressive strength test is described as a suggested method for determining UCS and E by ISRM [36] and ASTM-D-2938 [37]. In the current research, four prepared specimens were tested from each sample to determine UCS and E. The two mentioned parameters are calculated from the following equations:

$$UCS = \frac{4F}{\pi D^2} \qquad (2)$$

$$E = \frac{\Delta\sigma}{\Delta\varepsilon_a} \qquad (3)$$

In these equations, UCS is uniaxial compressive strength (MPa), F is maximum applied force upon the tested specimen (N), D is diameter of the tested cylindrical specimen (mm), E is secant modulus of elasticity (MPa), $\Delta\sigma$ is the change of applied stress upon the tested specimen (MPa), and $\Delta\varepsilon_a$ is the change of axial strength of the specimen during the test. Table 1 summarizes the average values of the obtained engineering properties of the tested rocks. The minimum and maximum values of UCS of the tested rocks are 11.66 and 38.49 MPa, respectively, which are moderate values based on ISRM [32]. The minimum and maximum values of E, $\gamma_d$, $n_e$, and $v_p$ are between 2.17 and 8.03 GPa, 22.01 and 25.78 kN/m$^3$, 0.37 and 7.55%, and 3759.79 and 5347.06 m/s, respectively. In accordance with Anon [38], the tested rocks have very low values of E, moderate to high values of $\gamma_d$, very low to moderate values of $n_e$, moderate to very

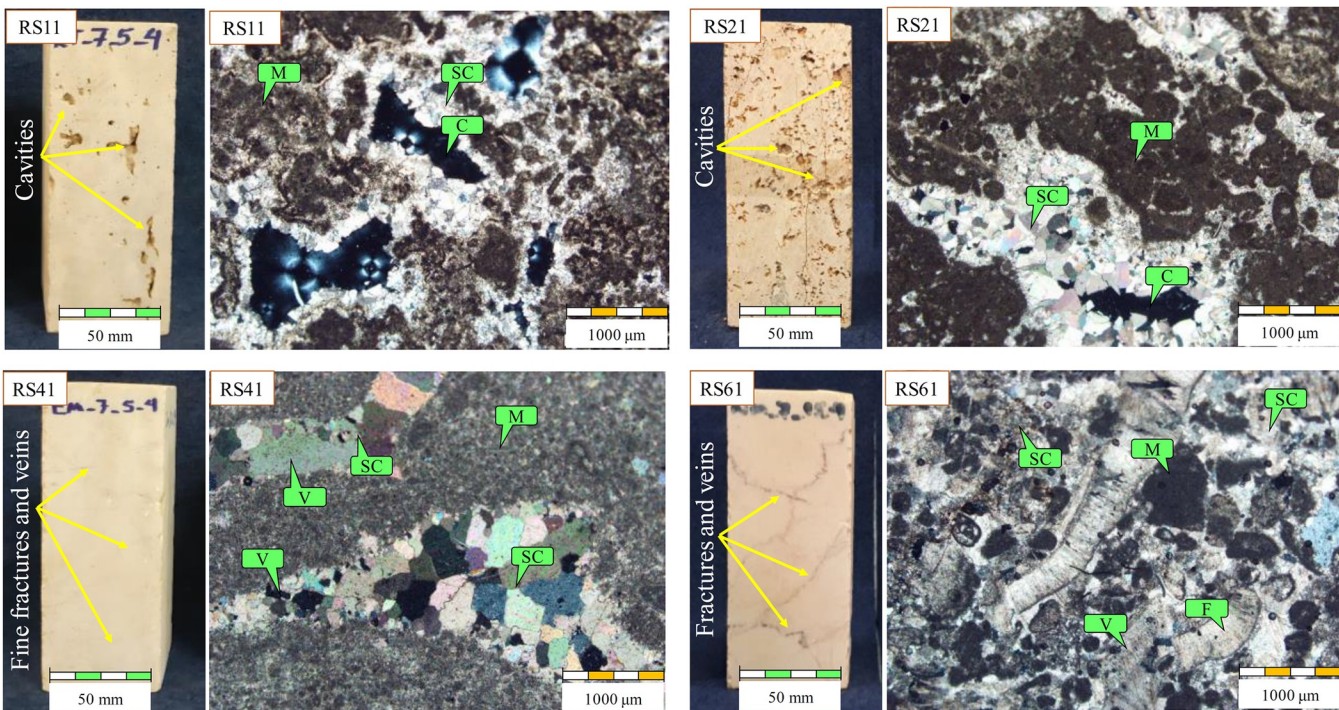

**Fig 2. Handy specimens and microscopic images of four tested rocks as representative samples.** (Note: SC: Sparry calcite, M: Micrite, C: Cavity, V: Vein, F: Fossil).

high values of $v_p$. The minimum and maximum values of HBN are 271.81 and 975.79 kgf/mm$^2$, respectively, which are high values based on ASTM E10-18 [35].

After calculating the required engineering characteristics of the rocks, distribution curve and histogram of the rock properties are provided which are presented in Fig 3, where Ishikawa formula is exploited for computing the number of bins [39].

The descriptive statistics of data including is also given in Table 2. The Pearson's product moment correlation coefficient of variables indicates the strength of the linear relationship between independent and dependent variables [40]. It is computed and is given in Fig 4.

## 3. Machine learning algorithms

Ensemble methods aggregate some models to enhance their generalizability and robustness. Two common ensemble methods are bagging and boosting applied to rock properties prediction in the literature [27, 41, 42], while stacking ensemble, a promising ensemble method, has received less attention in the related works. Compared to bagging and boosting, stacking ensemble method has three important characteristics; 1) it fully utilized training data, the model can be learned from all samples. 2) training in its first level is done by cross-validation, thus the trained model is robust and training overfitting phenomenon is not occurred. 3) it integrates different types of base learners; therefore, it takes their advantages and deals with their disadvantages. The last issue is so important since finding a suitable model for various datasets is so difficult. This section describes the stacking ensemble and base models exploited in this research.

### 3.1. Stacking ensemble learning

The stacking ensemble learning model was introduced by wolpert [43]. Recently, it has been successfully applied in various applications [44–50]. It combines some models together in

**Table 1. Engineering properties of the rocks.**

| Rock sample | Lithology | $\gamma_d$ (kN/m$^3$) | $n_e$ (%) | $v_p$ (m/s) | HBN (kgf/mm$^2$) | UCS (MPa) | E (GPa) |
|---|---|---|---|---|---|---|---|
| RS1 | Travertine | 24.07 | 3.14 | 4964.44 | 975.79 | 36.64 | 6.42 |
| RS2 | Travertine | 24.05 | 3.21 | 4905.59 | 762.06 | 30.72 | 5.70 |
| RS3 | Travertine | 24.03 | 3.37 | 4830.52 | 695.82 | 29.38 | 5.34 |
| RS4 | Travertine | 24.00 | 3.53 | 4645.79 | 637.86 | 28.22 | 5.29 |
| RS5 | Travertine | 24.27 | 3.48 | 4845.55 | 586.86 | 29.61 | 4.88 |
| RS6 | Travertine | 24.23 | 3.73 | 4690.47 | 563.62 | 27.24 | 4.70 |
| RS7 | Travertine | 24.21 | 3.95 | 4284.51 | 433.68 | 26.65 | 3.63 |
| RS8 | Travertine | 24.03 | 3.73 | 4738.30 | 433.68 | 26.71 | 3.95 |
| RS9 | Travertine | 23.95 | 4.06 | 4334.33 | 378.67 | 22.05 | 3.39 |
| RS10 | Travertine | 23.84 | 4.29 | 3876.24 | 366.56 | 21.13 | 2.78 |
| RS11 | Travertine | 23.49 | 3.16 | 4931.86 | 727.81 | 38.18 | 6.51 |
| RS12 | Travertine | 23.06 | 4.13 | 4818.39 | 611.56 | 29.01 | 6.13 |
| RS13 | Travertine | 22.94 | 4.88 | 4713.45 | 541.73 | 25.05 | 5.94 |
| RS14 | Travertine | 22.71 | 5.73 | 4621.48 | 483.21 | 24.17 | 5.30 |
| RS15 | Travertine | 22.53 | 4.39 | 4762.16 | 391.40 | 26.17 | 5.63 |
| RS16 | Travertine | 22.52 | 5.66 | 4640.35 | 355.01 | 23.58 | 4.53 |
| RS17 | Travertine | 22.49 | 6.66 | 4473.04 | 346.71 | 22.18 | 3.72 |
| RS18 | Travertine | 23.78 | 4.60 | 4498.85 | 344.00 | 23.89 | 5.15 |
| RS19 | Travertine | 23.70 | 6.40 | 4163.82 | 333.50 | 20.75 | 3.41 |
| RS20 | Travertine | 23.57 | 7.75 | 3770.74 | 271.81 | 19.18 | 2.84 |
| RS21 | Travertine | 22.58 | 4.19 | 4845.93 | 893.88 | 22.16 | 3.97 |
| RS22 | Travertine | 22.48 | 4.57 | 4794.15 | 762.06 | 19.52 | 3.75 |
| RS23 | Travertine | 22.15 | 5.07 | 4714.02 | 611.56 | 18.22 | 3.54 |
| RS24 | Travertine | 22.01 | 5.60 | 4578.96 | 586.86 | 17.85 | 3.41 |
| RS25 | Travertine | 23.27 | 5.04 | 4740.25 | 611.56 | 18.15 | 3.65 |
| RS26 | Travertine | 23.25 | 5.71 | 4639.86 | 586.86 | 16.94 | 3.31 |
| RS27 | Travertine | 23.21 | 6.54 | 4513.11 | 521.09 | 15.82 | 2.79 |
| RS28 | Travertine | 23.14 | 5.77 | 4671.77 | 483.21 | 17.38 | 3.11 |
| RS29 | Travertine | 23.04 | 6.65 | 4185.24 | 465.80 | 13.93 | 2.72 |
| RS30 | Travertine | 22.85 | 7.55 | 3759.79 | 449.31 | 11.66 | 2.17 |
| RS31 | Limestone | 25.22 | 0.63 | 5290.81 | 838.22 | 43.20 | 7.36 |
| RS32 | Limestone | 25.07 | 0.70 | 5116.82 | 695.82 | 30.83 | 6.21 |
| RS33 | Limestone | 24.95 | 0.78 | 4993.48 | 665.90 | 28.19 | 5.04 |
| RS34 | Limestone | 24.76 | 0.87 | 4859.59 | 637.86 | 23.81 | 4.43 |
| RS35 | Limestone | 25.29 | 0.74 | 5061.91 | 563.62 | 29.59 | 5.82 |
| RS36 | Limestone | 25.28 | 0.84 | 4880.37 | 483.21 | 25.41 | 4.65 |
| RS37 | Limestone | 25.25 | 0.93 | 4711.09 | 465.80 | 22.46 | 3.53 |
| RS38 | Limestone | 24.93 | 0.87 | 4645.05 | 449.31 | 24.36 | 5.47 |
| RS39 | Limestone | 24.85 | 0.99 | 4272.91 | 433.68 | 22.59 | 3.94 |
| RS40 | Limestone | 24.69 | 1.11 | 3919.26 | 378.67 | 19.67 | 3.03 |
| RS41 | Limestone | 25.10 | 0.44 | 5264.22 | 695.82 | 37.54 | 6.76 |
| RS42 | Limestone | 24.98 | 0.53 | 5148.04 | 541.73 | 27.31 | 5.52 |
| RS43 | Limestone | 24.83 | 0.63 | 5047.94 | 465.80 | 24.05 | 4.74 |
| RS44 | Limestone | 24.58 | 0.78 | 4928.61 | 418.86 | 22.19 | 3.92 |
| RS45 | Limestone | 24.68 | 0.74 | 5061.65 | 391.40 | 23.86 | 5.10 |
| RS46 | Limestone | 24.67 | 0.89 | 4888.37 | 378.67 | 21.77 | 3.82 |
| RS47 | Limestone | 24.64 | 1.00 | 4501.99 | 355.01 | 21.06 | 3.11 |

*(Continued)*

**Table 1.** (Continued)

| Rock sample | Lithology | $\gamma_d$ (kN/m$^3$) | $n_e$ (%) | $v_p$ (m/s) | HBN (kgf/mm$^2$) | UCS (MPa) | E (GPa) |
|---|---|---|---|---|---|---|---|
| RS48 | Limestone | 25.21 | 0.90 | 4810.27 | 344.00 | 19.61 | 4.64 |
| RS49 | Limestone | 25.14 | 1.21 | 4420.91 | 304.72 | 15.75 | 3.35 |
| RS50 | Limestone | 24.94 | 1.32 | 3987.41 | 295.95 | 15.30 | 2.66 |
| RS51 | Limestone | 25.43 | 0.15 | 5090.87 | 838.22 | 32.25 | 7.50 |
| RS52 | Limestone | 25.04 | 0.36 | 4959.56 | 762.06 | 29.18 | 6.53 |
| RS53 | Limestone | 25.01 | 0.37 | 4840.79 | 563.62 | 25.15 | 5.44 |
| RS54 | Limestone | 24.91 | 0.51 | 4591.70 | 541.73 | 24.84 | 4.55 |
| RS55 | Limestone | 25.16 | 0.43 | 4884.35 | 501.62 | 28.02 | 6.17 |
| RS56 | Limestone | 25.15 | 0.47 | 4677.25 | 465.80 | 23.10 | 5.00 |
| RS57 | Limestone | 25.12 | 0.62 | 4429.29 | 433.68 | 21.99 | 3.75 |
| RS58 | Limestone | 25.57 | 0.55 | 4854.55 | 418.86 | 26.15 | 5.71 |
| RS59 | Limestone | 25.48 | 0.60 | 4635.78 | 404.78 | 20.40 | 3.99 |
| RS60 | Limestone | 25.33 | 0.93 | 4262.11 | 344.00 | 19.22 | 2.67 |
| RS61 | Limestone | 25.40 | 0.37 | 5347.06 | 926.39 | 38.49 | 8.03 |
| RS62 | Limestone | 25.27 | 0.42 | 5173.13 | 762.06 | 32.19 | 6.79 |
| RS63 | Limestone | 25.13 | 0.49 | 5082.41 | 586.86 | 32.03 | 5.81 |
| RS64 | Limestone | 24.99 | 0.57 | 5013.99 | 541.73 | 29.93 | 4.43 |
| RS65 | Limestone | 25.78 | 0.44 | 5040.47 | 521.09 | 29.86 | 6.08 |
| RS66 | Limestone | 25.77 | 0.54 | 4861.91 | 501.62 | 27.34 | 4.62 |
| RS67 | Limestone | 25.74 | 0.61 | 4771.32 | 465.80 | 25.95 | 3.76 |
| RS68 | Limestone | 25.13 | 0.49 | 4912.53 | 404.78 | 26.54 | 5.51 |
| RS69 | Limestone | 25.04 | 0.63 | 4576.53 | 366.56 | 24.78 | 3.89 |
| RS70 | Limestone | 24.90 | 0.69 | 4229.14 | 344.00 | 22.29 | 3.03 |

order to enhance accuracy, generalization ability, and robustness. Stacking ensemble model consists of two levels, some base models are learned in the first level and one meta-learner is trained in the second level. Meta-learner's training data set is the altered version of original dataset and involves some synthetic features depend on base models' predictions. The training phase of stacking ensemble is illustrated in Fig 5. The models in the first level predicts the output variable in k-fold cross-validation manner, where one fold is considered as a validation set and the method is learned on other folds. The prediction process for first level is shown in Fig 6.

For optimizing base models' hyperparameters, the grid-search algorithm integrating with k-fold cross validation is applied. In the first step, the value range of hyperparameters are set, and then the model is trained by considering all combinations of hyperparameter values. Each combination is evaluated by computing the performance measure based on k-fold cross validation setting. Best combination is selected to train corresponding base model on all training data. These steps are shown in Fig 7.

After training base models on original data set, meta-learner is trained on synthetic data set. The details of stacking ensemble are given in Fig 8. To achieve suitable stacking ensemble model, base models should be diverse and have high performance. In this research, we design two stacking ensembles; one for predicting the UCS, and another for estimating the static E of carbonate rocks. Four classifiers are studied as base models, two base models for each stacking ensembles. The base models are SVR, RF, XGBoost, and MLP, since they have been successfully applied to the rock properties prediction [51]. SVR is a suitable method for dealing with nonlinear problems and has been achieved to high quality results when available data are rare [52]. XGBoost outperformed other machine learning methods in many challenges [53]. MLP is a well-known and powerful learning method to resolve challenges in the rock data, and RF

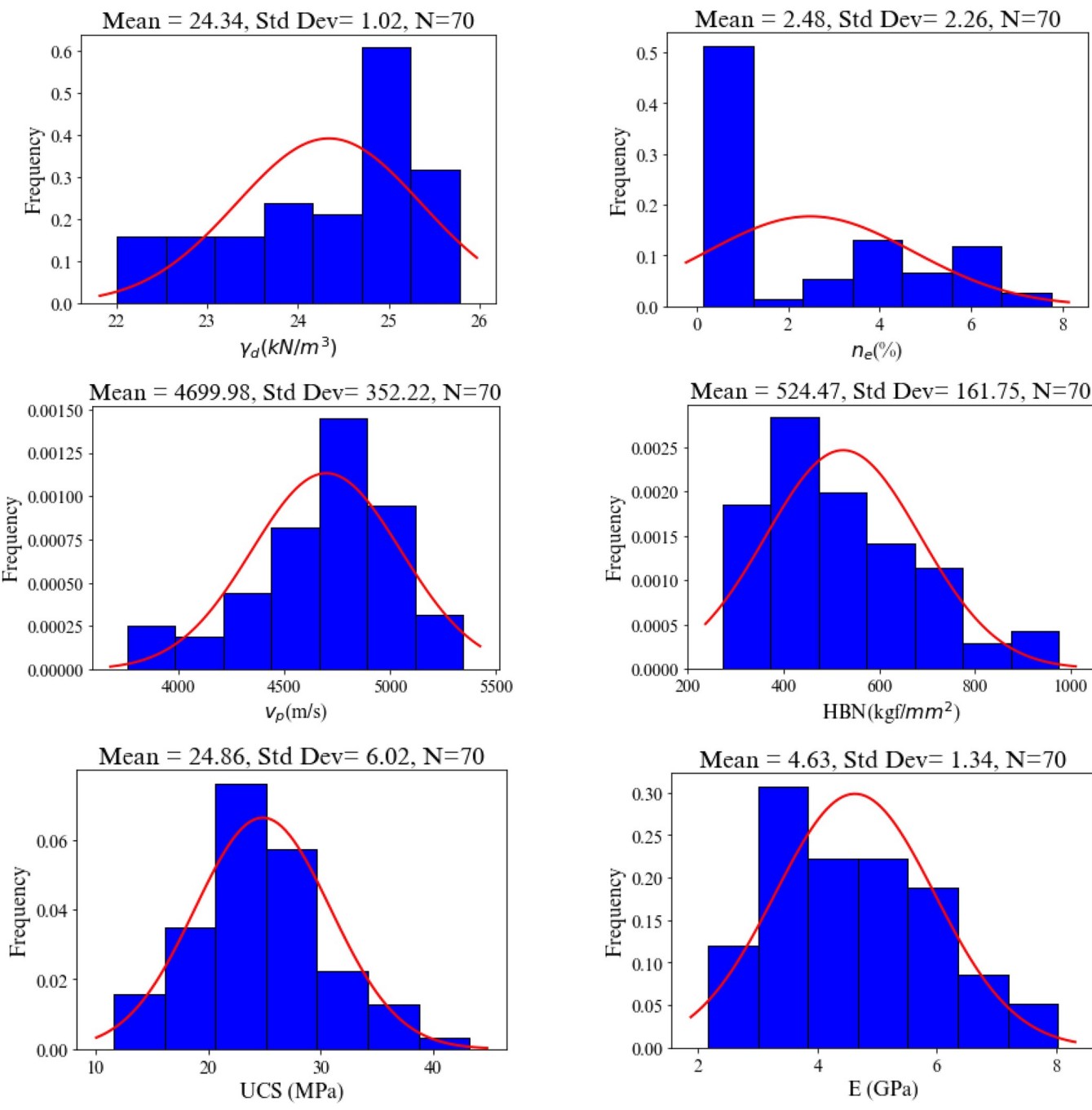

**Fig 3. Distribution curves and histograms of the rock properties in the database.**

frequently applied to rock properties prediction in the literature [41, 54–56]. These methods will be summarily described in next sections.

### 3.2. Support vector regression

The SVR is a prominent model for predicting dependent variable employing statistical learning theory. It is based on the structural risk minimization principal and exploits the kernel

**Table 2. Descriptive statistics of data set.**

| Statistics | E (GPa) | UCS (MPa) | $\gamma_d$ (kN/m³) | $N_e$ (%) | $v_p$ (m/s) | HBN (kgf/mm²) |
|---|---|---|---|---|---|---|
| Mean | 4.63 | 24.86 | 24.34 | 2.48 | 4699.98 | 524.47 |
| St. Dev. | 1.35 | 6.07 | 1.02 | 2.27 | 354.76 | 162.92 |
| Minimum | 2.17 | 11.66 | 22.01 | 0.15 | 3759.79 | 271.81 |
| Maximum | 8.03 | 43.20 | 25.78 | 7.75 | 5347.06 | 975.79 |
| Skewness | 0.35 | 0.53 | -0.66 | 0.71 | -0.78 | 0.80 |
| Kurtosis | -0.64 | 0.53 | -0.78 | -0.93 | 0.37 | 0.03 |

trick. By considering $m$ training samples $\{x_i, y_i\}_{i=1}^{m}$, where $x_i = <x_i^1, \ldots x_i^D> \in \mathbb{R}^D$ and $y_i \in \mathbb{R}$, SVR estimates output variable by the following equation:

$$f(x) = <\omega, \phi(x)> +b \tag{4}$$

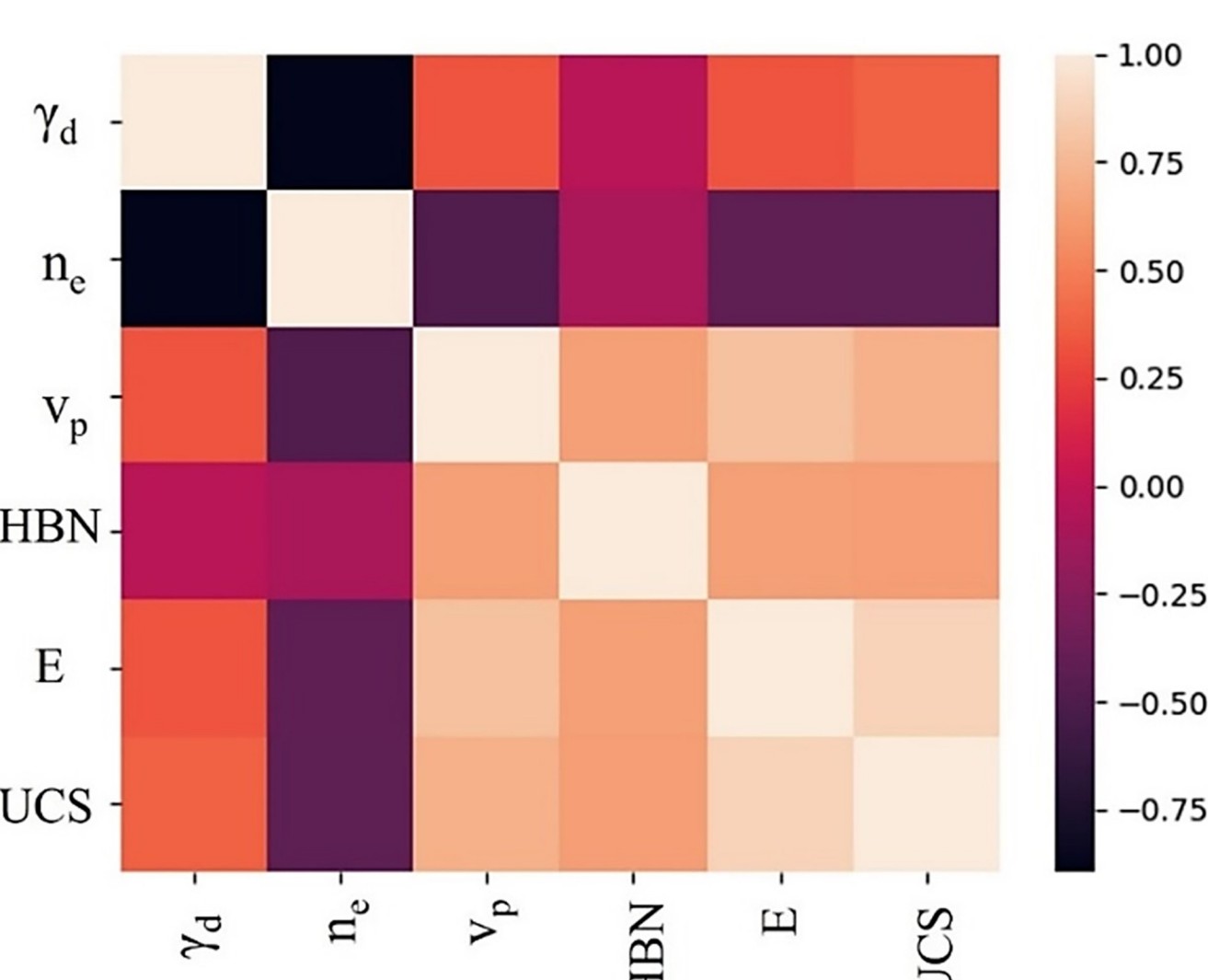

**Fig 4. Pearson's correlation coefficient for data.**

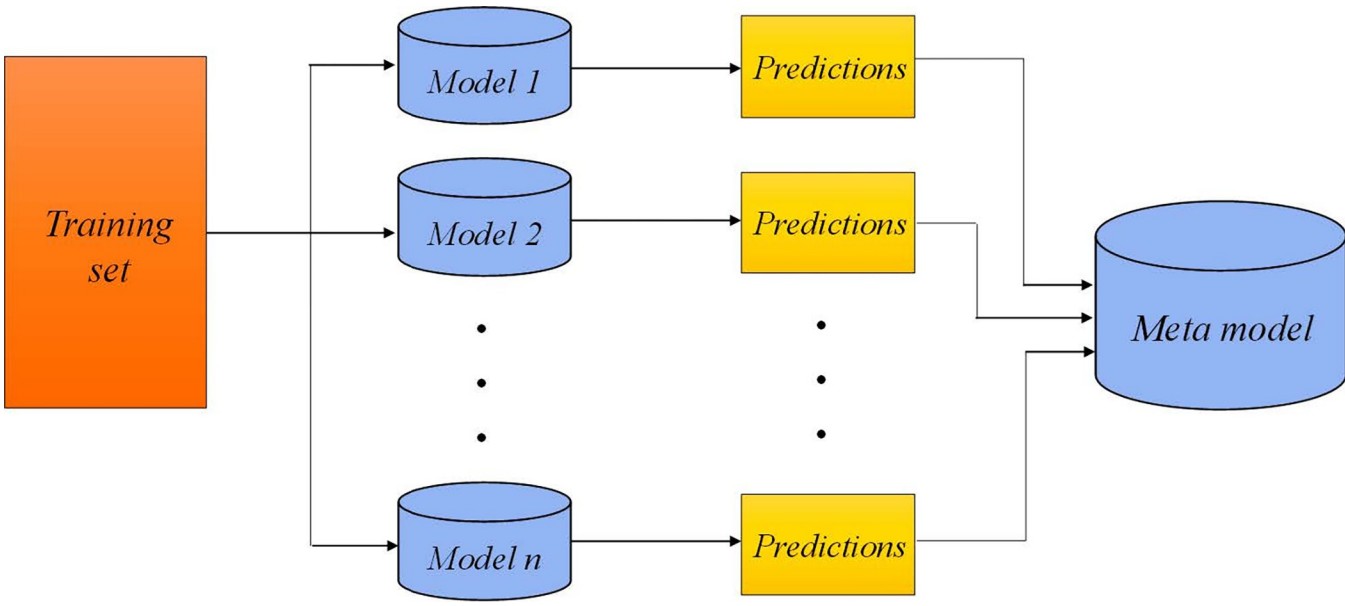

**Fig 5. Training phase of the stacking model.**

where $b$ is intercept, $\omega \in \mathbb{R}^{d_k}$ is the weight feature vector, and $\phi(.) : \mathbb{R}^D \to \mathbb{R}^{d_k}$ is a mapping from the input space to high dimensional new space, and $d_k$ is the dimension of feature space that is implicitly defined. The notation of $<.,.>$ indicate the dot product [57]. A kernel function is commonly employed in SVR to transform input data to a feature space with high dimensional for considering data non-linearity. Radial Basis Function (RBF) is the most widely used kernel function that computes the similarity of $x_i$ and $x_j$ by $\exp(-\gamma\|x_i - x_j\|^2)$, where $\gamma$ is its parameter.

The optimization problem of SVR consists of two terms of regularization, and loss function as follow:

$$\min \frac{1}{2}\|\omega\|^2 + C\sum_{i=1}^{m}(\xi_i + \xi_i^*) \tag{5}$$

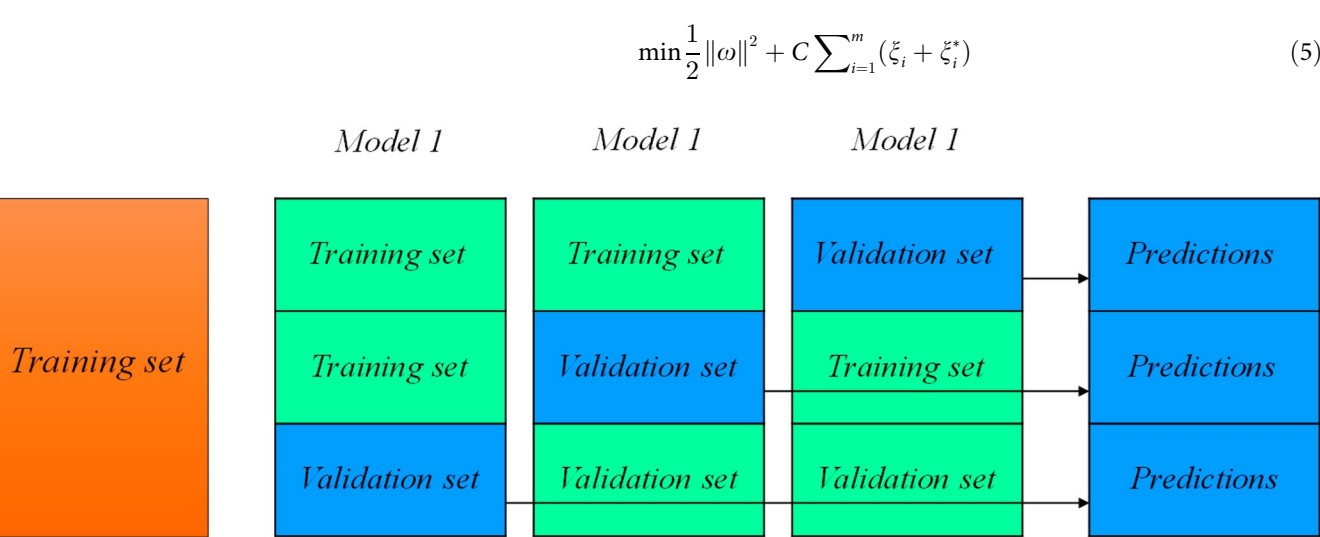

**Fig 6. Predictions process performed by a model in the first level of the stacking ensemble.**

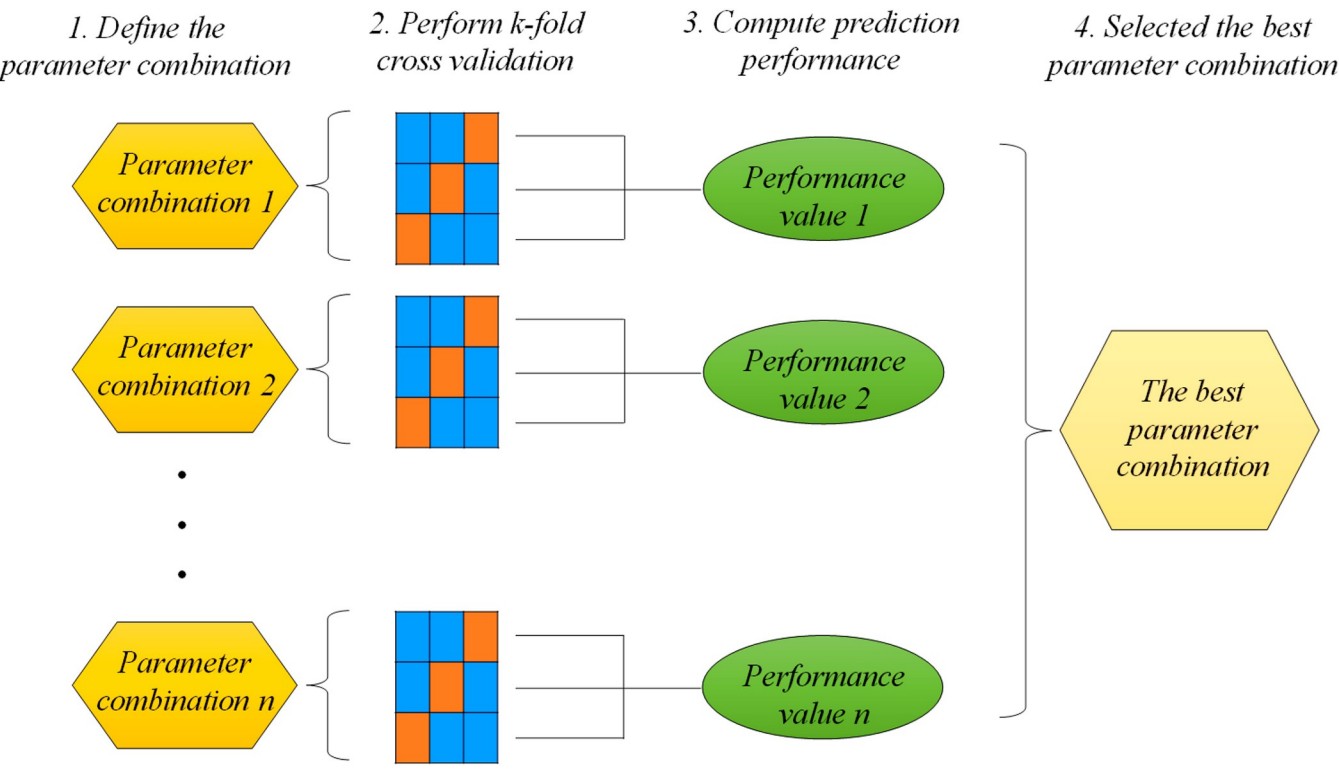

**Fig 7. Grid search with 3-fold cross validation.**

$$\text{such that} \begin{cases} \forall i, \quad 1 \leq i \leq m : & y_i - <\omega, \phi(x_i)> -b \leq \varepsilon + \xi_i \\ \forall i, \quad 1 \leq i \leq m : & <\omega, \phi(x_i)> +b - y_i \leq \varepsilon + \xi_i^* \\ \forall i, \quad 1 \leq i \leq m : & \xi_i, \xi_i^* \geq 0 \end{cases}$$

where $l^2$ −norm and $\varepsilon$-insensitive is applied as loss function and regularization terms, respectively, and $C>0$ controls the trade-off between two terms. $\xi$ and $\xi^*$ are slack variables that are introduce for tolerating misclassification in training data. For more details on SVR refer to Smola [57]. SVR's various variants have been successfully applied for predicting engineering properties of rocks [58, 59].

## 3.3. Random forest

The random forest is an ensemble of classification and regression trees (CART) with suitable characteristics including 1) supporting nonlinearity, 2) being non-parametric, 3) fast training, 4) random subspace, and 5) being resistance to overfitting. The base tree models are built on randomly selected input features of randomly samples of original data set by bagging manner. This randomness increases the diversity of the base trees. For building each tree, input space partitions to two parts in the recursive manner in order to data in each leaf be pure as much as possible. The purity in the regression task is commonly defined by mean squared error measure, where a regression model is learned from data in each leaf. The predicted value in the regression task is computed by averaging forecasting values of each base learner. The RF has been recently applied for various applications including the prediction of engineering properties of rocks [41, 54, 55]. The number of base trees and maximum depth of them are two

**Algorithm**   Stacking Ensemble

**Input:** Original training dataset $D$, first-level algorithms $L_1, L_2, \ldots, L_l$, second level algorithm, $L_s$

**Output:** base models, $\{M_1, \ldots, M_l\}$, met-model, $M_s$

1) Train $M_i$ (i$\in \{1, \ldots, l\}$): Utilize a k-fold cross-validation method to divide the training dataset $D$ into k datasets with almost equally sized, $\{D_1, \ldots, D_k\}$, one of them is used as a test dataset, while the remaining (k-1) subsets are used as training datasets of the current first layer and a classifier is trained based on the algorithm $L_i$. This setting is repeated for all possible combinations of input parameters, and the best parameters are selected based on pre-specified performance measure. $M_i$ is trained on all data set with resulting parameters.

2) Train $M_s$: Use the target prediction results of base learners on training data in the combination of target values of the original training data as the input training dataset of the second layer meta-model. Then, train $M_s$ according to algorithm $L_s$ exploiting this training data set.

**Fig 8. Pseudocode of building models in the stacking ensemble.**

effective hyperparameters in the RF. Restricting tree depths in the training phase can enhance the generalization ability of them and reduce the memory usage.

### 3.4. XGBoost

The XGBoost is an effective scalable method that combines some trees in an iterative boosting manner. It learns some tree models like the RF, but it differs from the RF in the training details. $i$th predicted value $\hat{y}_i$ is computed as

$$\hat{y}_i = \phi(\mathbf{x}_i) = \sum_{k=1}^{K} f_k(x_i), f_k \in \mathcal{F} \tag{6}$$

where $f_k$ is an independent tree and $\mathcal{F} = \{f(x) = \omega_{q(x)}\}(q : \mathbb{R}^D \to T, \omega \in \mathbb{R}^T)$ indicates the space of regression trees

$$\mathcal{L}(\phi) = \sum_i l(\hat{y}_i, y_i) + \sum_k \Omega(f_k) \tag{7}$$

where $\Omega(f) = \alpha T + \frac{1}{2}\beta\|\omega\|^2$, $l$ is a convex loss function that vanishes the violation of precited values $\hat{y}_i$ from target value $y_i$, $\Omega$ is the regularization term that prevent from overfitting and

consists of two terms: the number of leaves, *T*, and the L2-norm of $\omega$ controlling the complexity of the model. The optimization of Eq 7 is done in the additive manner. For studying more details about it, please refer to Chen [60]. Three important parameters of the XGBoost are the number of trees, maximum tree depth, and learning rate. Since the XGBoost learns each tree to correct the error of the existing sequence of trees, it is subject to overfit. To prevent overfitting, a weight factor is assigned to the correction made by each new tree. This weight factor is learning rate.

### 3.5. MLP

A Neural network is a powerful model for representing both linear and non-linear relations between inputs and outputs. The MLP is a most common neural network that it exploits back propagation algorithm for learning its weights. It is a supervised and feedforward network. Training phase of the MLP includes two steps: first, the selection of neural network architecture, and second, adjusting connections' weights. A typical MLP has an input layer and an output layer. It may be had one or more hidden layers that extract some important features of input data and are not directly accessible. Some neurons are included in each layer and an activation function is assigned to each neuron. The MLP can learn nonlinear patterns from data. Two well-known weight optimization methods in the MLP are L-BFGS (Limited-Memory Broyden-Fletcher-Goldfarb-Shanno) [61] and Adam (adaptive moment estimation) [62]. The L-BFGS belongs to a class of Quasi-Newton methods and the Adam is an efficient stochastic optimization method that only needs first-order gradients plus little memory requirement.

## 4. Model development, results and discussions

This part explains the application of the stacking ensemble model for estimating the UCS and static E of carbonate rocks. Section 3.1 explains some settings included in our experiments, section 3.2 describes the performance measures that exploited for assessing the proposed methods, and finally Section 3.3 gives the results and discuss about them.

### 4.1. Settings

The dataset is spitted randomly to training and test, where 20% of data is considered as test and 80% of data form training data. The input data is normalized before applying machine learning methods to scale all features in the range of [0,1]. This step is so important to eliminate the effect of varying ranges of various features. This normalization is done by the following relation:

$$z_i^d = \frac{x_i^d - x_{\min}^d}{x_{max}^d - x_{min}^d}, (1 \leq i \leq m; 1 \leq d \leq D) \tag{8}$$

where $x_i^d$ is the value of *d*th input variable of *i*th data, $x_{\min}^d$ and $x_{max}^d$ are minimum and maximum values of *d*th input variable, $z_i^d$ is the scaled value of $x_i^d$, and D is 4 in our data which equals the number of independent variables including $\gamma_d, n_e, v_p$, and *HBN*.

### 4.2. Performance measures

In this paper, the performance of the estimation is presented in the terms of the coefficient of determination ($R^2$), Root Mean Squared Error (RMSE), Mean Squared Error (MSE), the Mean Absolute Error (MAE), Variance Accounted For (VAF), Index of Scatter (IOS), agreement index (IOA), Mean absolute percentage error (MAPE), Weighted Mean absolute percentage error (WMAPE), Performance Index (PI), and a20index are computed. These metrics are

frequently employed to assess regression issues [63, 64]. RMSE, MSE, MAE, MAPE, WMAPE, and IOS indicated the error prediction. $R^2$ specifies the appropriateness of the fitted model and is in the range of [0,1] and larger $R^2$ values and smaller MSE, MAE, RMSE, MAPE, WMAPE, and IOS values indicate better performance. The measure of a20index indicates the quantity of samples that correspond to the observed values within the margin of ±20% deviation, as identified by m20, in relation to the predicted values. A higher a20index value specifies better predictive accuracy. The relations of these measures are given as follows:

$$RMSE = \sqrt{\frac{\sum_{i=1}^{n}(y_i - \hat{y}_i)^2}{m}} \tag{9}$$

$$MAE = \frac{1}{m}\sum_{i=1}^{N}|y_i - \hat{y}_i| \tag{10}$$

$$R^2 = 1 - \frac{\sum_i(y_i - \hat{y}_i)^2}{\sum_i(y_i - \bar{y})^2} \tag{11}$$

$$MSE = \frac{\sum_{i=1}^{n}(y_i - \hat{y}_i)^2}{m} \tag{12}$$

$$VAF = \left(1 - \frac{var(y_i - \hat{y}_i)}{var(y_i)}\right) \times 100 \tag{13}$$

$$IOS = \frac{RMSE}{Average\ of\ actual\ values} \tag{14}$$

$$IOA = 1 - \frac{\sum_{i=1}^{n}(\hat{y}_i - y_i)}{\sum_{i=1}^{n}(y_i - Average\ of\ actual\ values)} \tag{15}$$

$$MAPE = \frac{1}{n}\sum_{i=1}^{n}|\frac{\hat{y}_i - y_i}{\hat{y}_i}| \times 100 \tag{16}$$

$$WMAPE = \frac{\sum_{i=1}^{n}|\frac{\hat{y}_i - y_i}{\hat{y}_i}| \times \hat{y}_i}{\sum_{i=1}^{n}\hat{y}_i} \tag{17}$$

$$PI = R^2 + \left(\frac{VAF}{100}\right) - RMSE \tag{18}$$

$$a20 - index = \frac{m20}{n}$$

## 4.3. Prediction performances

**a. Tunning hyperparameters of base models.**    In the first stage, we train some state-of-the-art machine learning methods consisting of XGBoost, RF, SVR and MLP. Grid search in the combination of k-fold cross validation is applied for parameter optimization of all base models. Mean of MSE is computed in each parameter configurations of grid search, and the

**Table 3. Parameter setup for each model.**

| Method | Tunning Parameter | Parameter Range |
|---|---|---|
| XGBoost | The number of estimators | {400,600,800,1000, 1200,1400,1600,1800, 2000} |
| | Maximum tree depth | {5,10, 20,30,40,50, 60, 70, 80, 90, 100} |
| | Learning rate | {0.01, 0.1, 0.2, 0.3,0.4, 0.5,0.6,0.7,0.8,0.9} |
| MLP | The number of hidden layer neurons | {7,8,9,10,11,12,13, 14, 15, 16, 17, 18, 19, 20} |
| | The solver of weight optimization | {'L-BFGS, 'Adam'} |
| SVR | $C$ | {0.001, 0.01, 0.1, 1, 10} |
| | $\gamma$ | {0.001, 0.01, 0.1, 1, 10, 100} |
| | $\varepsilon$ | {0.001, 0.01, 0.1, 1, 10} |
| RF | The number of trees in the forest | {200, 300, 400, 500} |
| | Maximum tree depth | {None,1,2,3,4,5,6,7,8,9,10} |

best result on the validation data indicates the best parameter settings. The optimized parameters and their ranges are given in Table 3.

A MLP model with one hidden layer also is considered in our experiments as a base model. The increasing hidden layer may reduce the error of the model, but also increases both network complexity and training time and leads overfitting. The influence of the number of hidden layer neurons and the method of weight optimization are illustrated in Fig 9. As demonstrated in this figure, for estimating the UCS, the number of neurons is more effective than the E, and the number of neurons and weight optimization method were set to 14 and L-BFGS, respectively. For predicting the E, Adam weight optimization function concludes more stable result than L-BFGS. The number of neurons by considering this weight optimization function does not have many impacts on MSE, our experiments select 17 neurons.

For the base model of the random forest, the effect of parameters is shown in Fig 10. The number of estimators is not so effective on the MSE for predicting E, and with maximum tree depth less than 3 overfitting is not occurred, based on the figure. The number of 200 trees with maximum depth of 3 are best configuration for predicting E. Similar manner exists for predicting UCS, where 400 trees with maximum of depth of 6 are found by grid search.

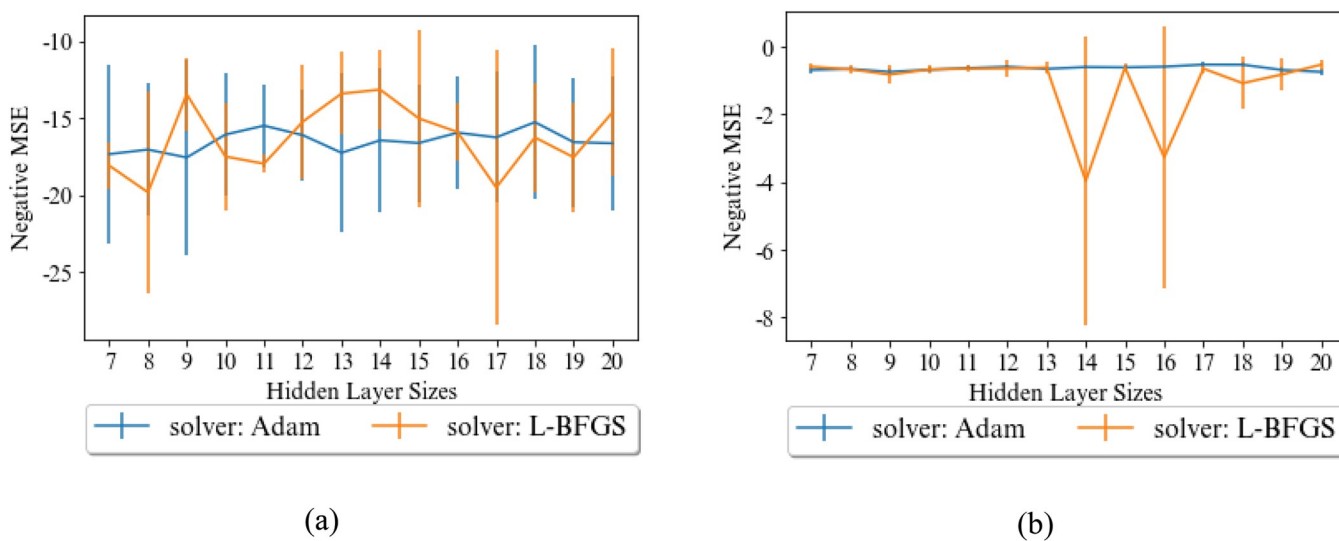

(a) (b)

**Fig 9.** Effects of the number of neurons in hidden layer and the solver of weight optimization in the MLP on MSE of predicting a) UCS, and b) E.

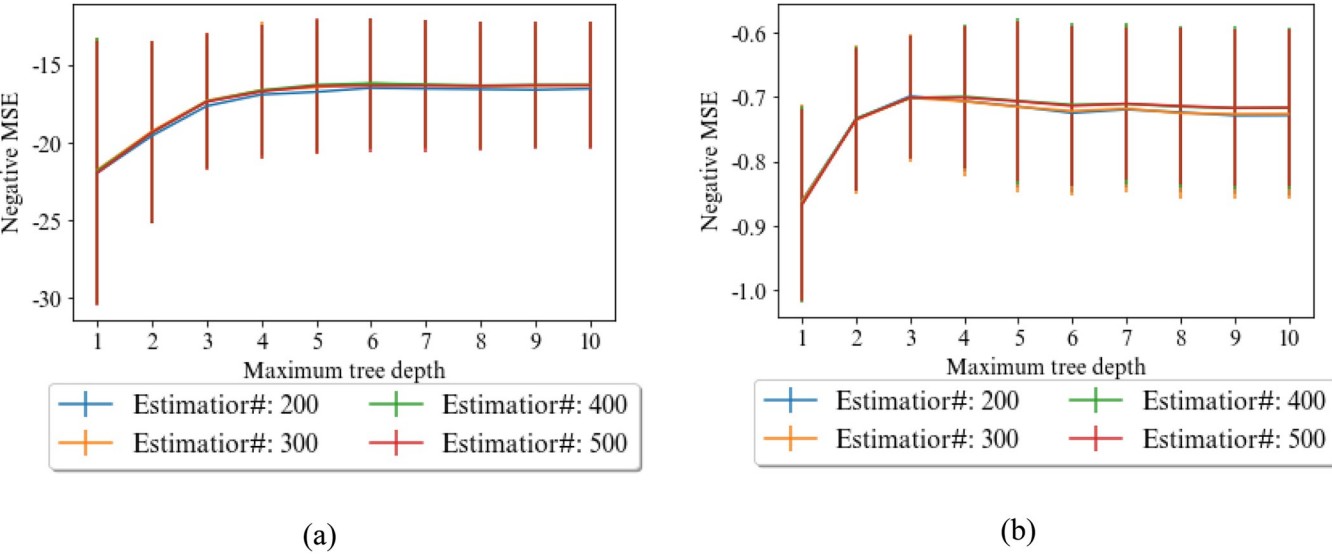

**Fig 10.** Effects of parameters of RF on MSE of forecasting a) UCS, and b) E.

(a)

(b)

**Fig 11.** Effect of parameters of the SVR on MSE of predicting a) UCS, and b) E.

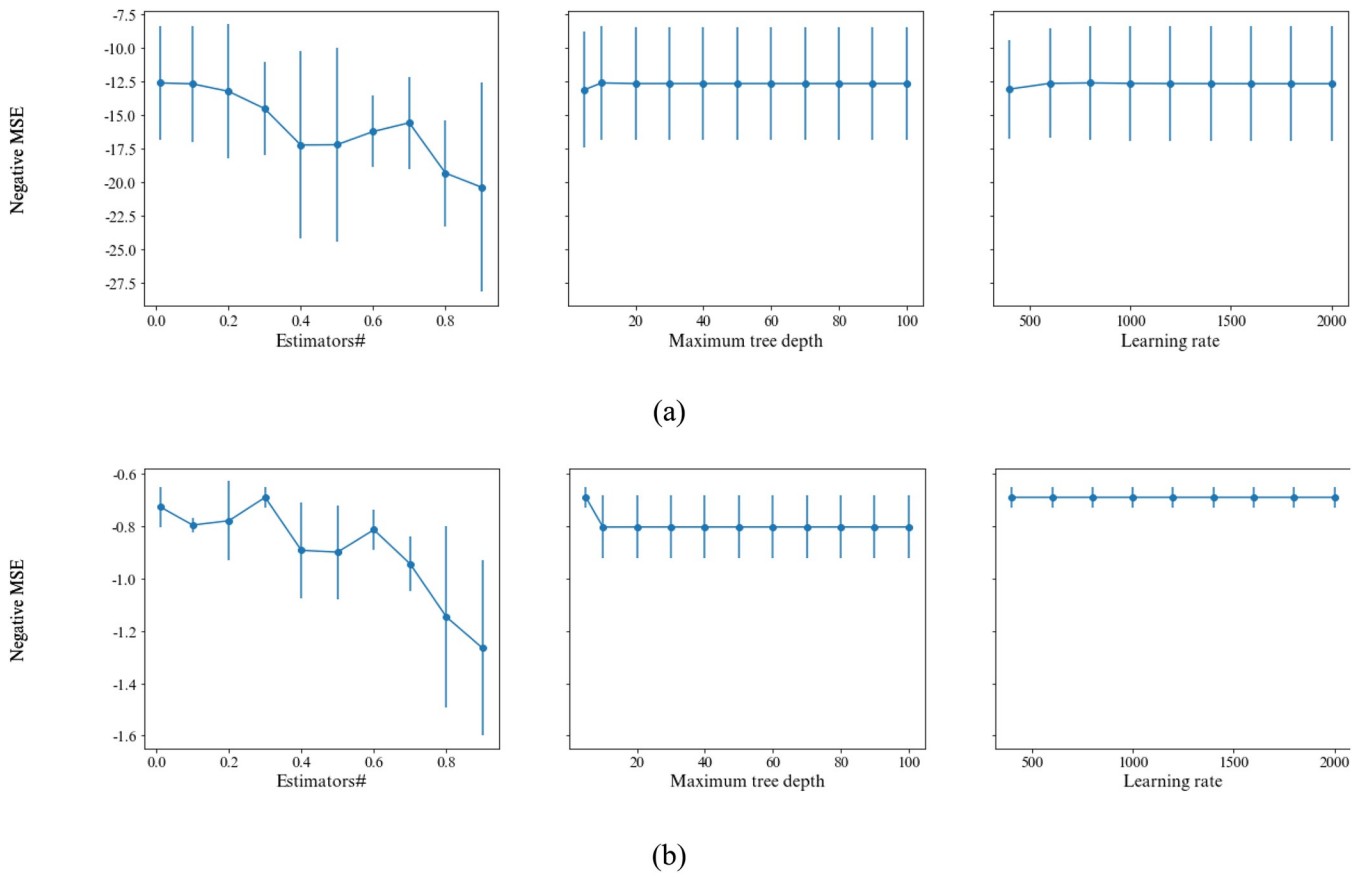

**Fig 12.** Effects of parameters of the XGBoost on MSE of predicting a) UCS, and b) E.

The impact of the SVR parameters on the MSE is also illustrated in Fig 11. In both parts of this figure, two parameters are fixed to their best values and only the mean of negative of mean squared error is plotted for another parameter. As it shown, the SVR are sensitive to all parameters. C = 10 and $\gamma$ = 1 are the best parameter settings for predicting both of UCS and E. The value of $\varepsilon$ is set to 1 and 0.1 for predicting UCS and E, respectively.

The influence of the XGBoost's parameters is also assessed and the results are given in Fig 12. Similar to the SVR parameters, one parameter value is changed in each time and other parameter values are set to their best values. The suitable values of these parameters for predicting E got achieved to 0.3, 5, and 400, respectively. The increasing values of all parameters leads to high error prediction; however, this effect is observed more strongly in the number of predictors. Increasing the number of predictors increases the model complexity and causes overfitting. Similar trend is maintained in predicting UCS, while the best values are 0.01, 10, and 800 for learning rate, maximum of depth, and the number of tree predictors.

**b. Testing the stacking ensemble model.** We conducted two stacking ensembles; one for predicting E by exploiting the SVR and XGBoost as base models, and another stacking ensemble model for predicting the UCS by using the RF and MLP as the first stage's models. The meta-learners are the SVM and MLP for predicting the E and UCS, respectively.

After hyperparameter tunning of base models, the hyperparameters of meta-learners are also tuned by grid search and then, two stacking ensemble models were trained for predicting the E and UCS. The range of parameters of meta-learners is also according to values given in

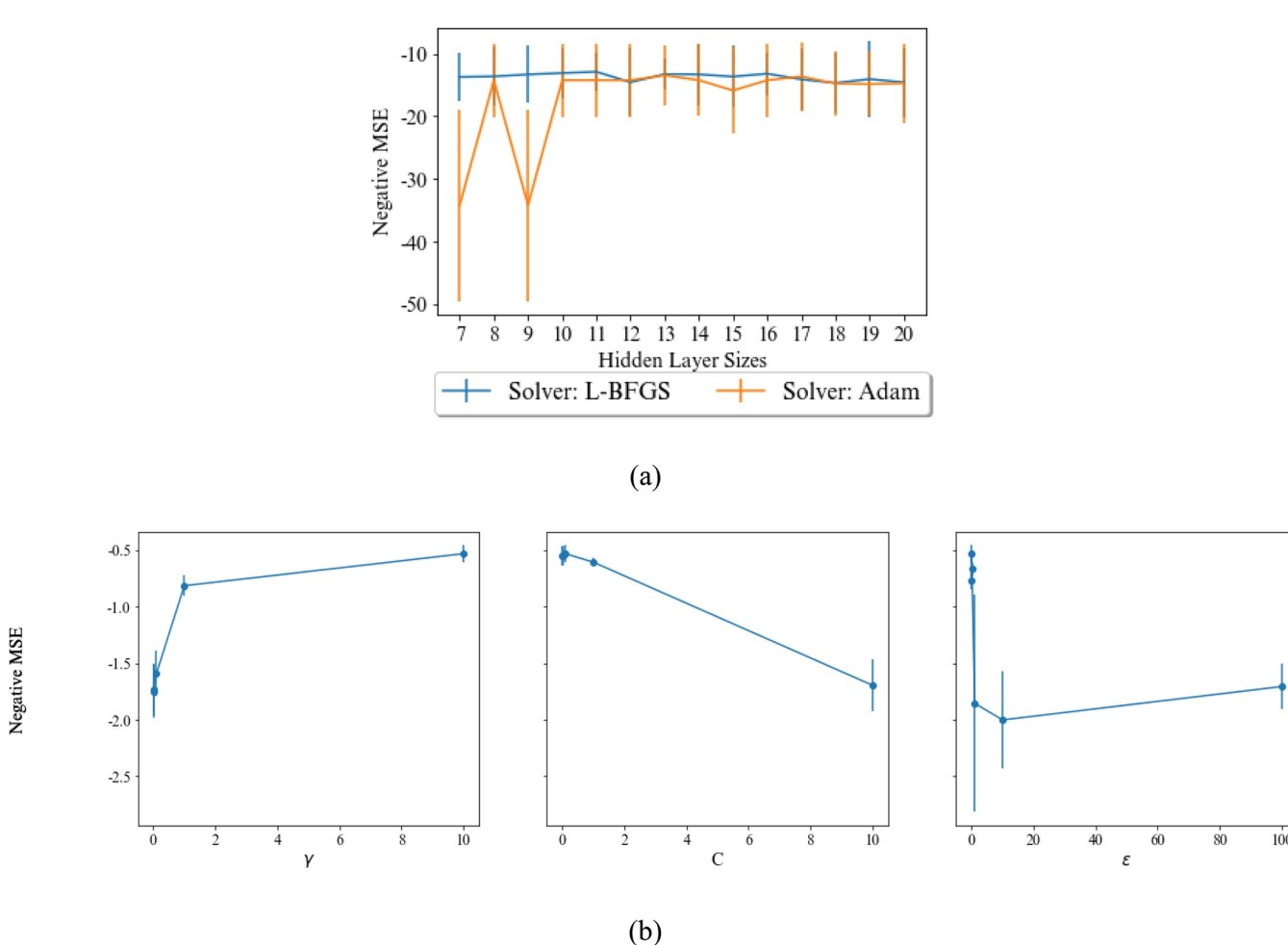

**Fig 13.** Effects of parameters of a) SVR, and b) MLP as the meta-learner for predicting the UCS, and E, respectively.

Table 3. The results of grid search are illustrated in Fig 13. C = 10, epsilon = 0.1, and gamma = 0.01 are the best values of the SVR parameter. The number of neurons in the MLP's hidden layer is also set to be 11.

The obtained performance results for predicting the UCS by four base models and the proposed stacking ensemble are also listed in Table 4. The suitability of the proposed model was confirmed with MAE, MSE, RMSE, $R^2$, IOA, and IOS. The results confirm that the stacking ensemble is superior on all base models.

*Table 5 gives the obtained performance metrics of the developed stacking ensemble and four base learners applied for predicting the E on the testing data. It is clear from the obtained results*

**Table 4. Performance results for predicting the UCS (MPa) in the testing phase.**

| Method | MAE | MSE | RMSE | R² | VAF | MAPE | PI | a20-index | WMAPE | IOA | IOS |
|---|---|---|---|---|---|---|---|---|---|---|---|
| XGBoost | 2.155 | 8.672 | 2.945 | 0.796 | 82.502 | 0.115 | 0.856 | 0.857 | 0.097 | 1.000 | 0.121 |
| MLP | 1.665 | 4.170 | 2.042 | 0.902 | 63.284 | 0.185 | 0.682 | 0.714 | 0.149 | 1.000 | 0.188 |
| SVR | 2.184 | 8.388 | 2.896 | 0.803 | 84.804 | 0.116 | 0.872 | 0.857 | 0.091 | 1.000 | 0.120 |
| RF | 2.324 | 7.903 | 2.811 | 0.814 | 83.191 | 0.120 | 0.870 | 0.857 | 0.097 | 1.000 | 0.117 |
| Stacking Ensemble | 1.657 | 3.867 | 1.967 | 0.909 | 74.531 | 0.149 | 0.785 | 0.714 | 0.120 | 1.000 | 0.006 |

**Table 5. Performance results for predicting the E (GPa) in the testing phase.**

| Method | MAE | MSE | RMSE | R² | VAF | MAPE | PI | a20-index | WMAPE | IOA | IOS |
|---|---|---|---|---|---|---|---|---|---|---|---|
| XGBoost | 0.631 | 0.544 | 0.738 | 0.762 | 69.262 | 0.237 | 0.721 | 0.643 | 0.196 | 1.000 | 0.229 |
| MLP | 0.610 | 0.597 | 0.773 | 0.739 | 69.180 | 0.191 | 0.786 | 0.643 | 0.173 | 1.000 | 0.210 |
| SVR | 0.562 | 0.494 | 0.703 | 0.784 | 80.098 | 0.147 | 0.860 | 0.714 | 0.127 | 1.000 | 0.160 |
| RF | 0.710 | 0.718 | 0.847 | 0.686 | 73.018 | 0.185 | 0.776 | 0.714 | 0.161 | 1.000 | 0.192 |
| Stacking Ensemble | 0.483 | 0.386 | 0.621 | 0.831 | 84.447 | 0.130 | 0.870 | 0.786 | 0.115 | 1.000 | 0.029 |

*that the stacking ensemble out performs base models in the terms of MSE, MAE, RMSE, $R^2$, VAF, MAPE, PI, a20_index, WMAPE, and IOS.*

For more clarity, the predicted and experimental values of the UCS and E by various models studied in this research in both training and testing phase are displayed in Figs 14 and 15, respectively. Furthermore, the scatter plot of actual and predicted values by four base models and stacking ensemble model are shown in Figs 16 and 17. The correlation of 0.83 and 0.91 confirm the suitability of the proposed method in the comparison of the base models. The results show that the predicted values in stacking ensembles are closer to observed valued than the output of other models.

For more Analysis, the training time of the studied methods are also compared in Table 6. The reported training time was obtained from a workstation equipped with an Intel(R) Xeon (R) W-2150B CPU operating at a frequency of 3 GHz, complemented by 64G of RAM.

## 4.4. Sensitivity analysis

Sensitivity analysis assesses the efficacy of input parameters in forecasting the output parameter(s). This examination serves to ascertain the parameter that exerts the greatest influence on the prediction. Moreover, it can be employed to construct the most optimal AI models by disregarding inconsequential input parameters. These analyses of sensitivity may take on either a

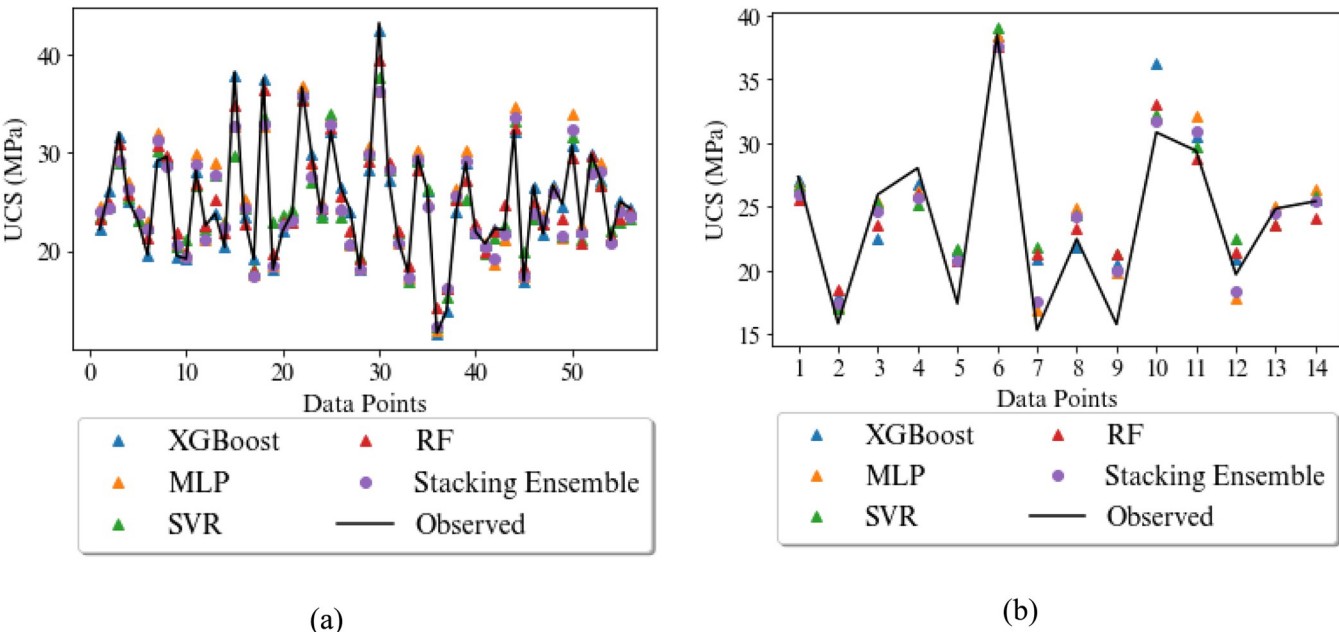

(a)                                                                          (b)

**Fig 14.** Measured and predicted UCS (MPa) for the rocks by the XGBoost, MLP, SVR, RF and stacking ensemble in the a) training phase and b) testing phase.

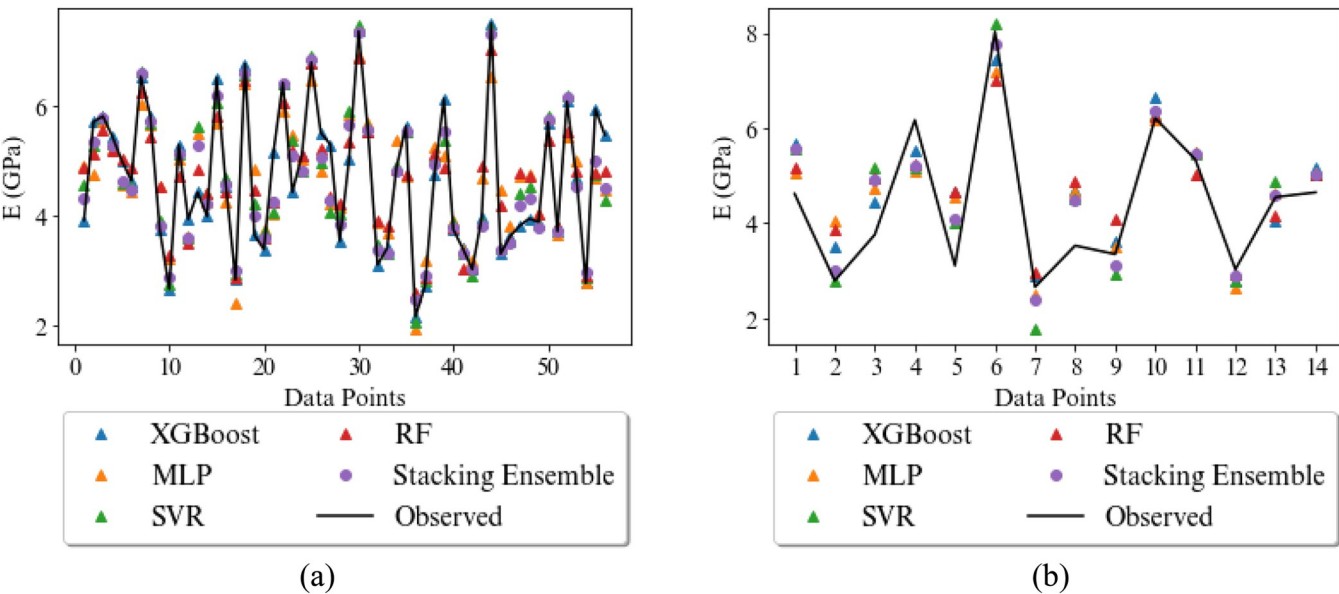

**Fig 15.** Measured and predicted E (GPa) for the rocks by the XGBoost, MLP, SVR, RF and stacking ensemble in the a) training phase and b) testing phase.

linear or nonlinear form. Numerous scholars have employed the cosine amplitude method (CAM) in conducting sensitivity analysis [64, 65]. CAM determines the sensitivity of input variables using below equation:

$$CAM = \frac{\sum_{k=1}^{m} x_{ik} x_{jk}}{\sqrt{\sum_{k=1}^{m} x_{ik}^2 \sum_{k=1}^{m} x_{jk}^2}} \tag{19}$$

where $X_i$ is the $i$th input, and $X_j$ I the $j$th output. The computed CAM for four input variables with UCS (MPa) and E (GPa) are illustrated in Fig 18. As it shown, $n_e$ has the lowest influence on both two outputs and other three input parameters have relatively high impact.

## 4.5. Discussion

In spite of high generalization ability and good accuracy of the studied base models in this research, applying them on prediction of the rock properties have some imperfections. The instances may be highly nonlinear and the SVR exploits only once from kernel function to transform the sample data to the high-dimensional feature space, while a single mapping cannot guarantee finding the optimal separable feature space. Exploiting MLP only in one level can lead to getting stuck in local optimum. The RF and XGBoost are based on bagging and boosting, respectively and suffer from their limitations. Both of them are the ensemble of some trees and don't exploit the advantages of other models.

In the current study, it has been verified that the combination of certain models within the stacking ensemble framework can enhance the achieved results according to various metrics. The proposed approach exhibits significant improvements in RMSE, MSE, MAE, VAF, IOS, MAPE, WMAPE, PI, and a20index when estimating E. This suggests that the proposed method can serve as a viable alternative to other methods for estimating E, as it exhibits low error and high explanatory power in relation to the variance in E. Moreover, the stacking ensemble method also enhances the measures of RMSE, MSE, MAE, and IOS when predicting UCS. This indicates that the proposed method effectively captures the variability in UCS and

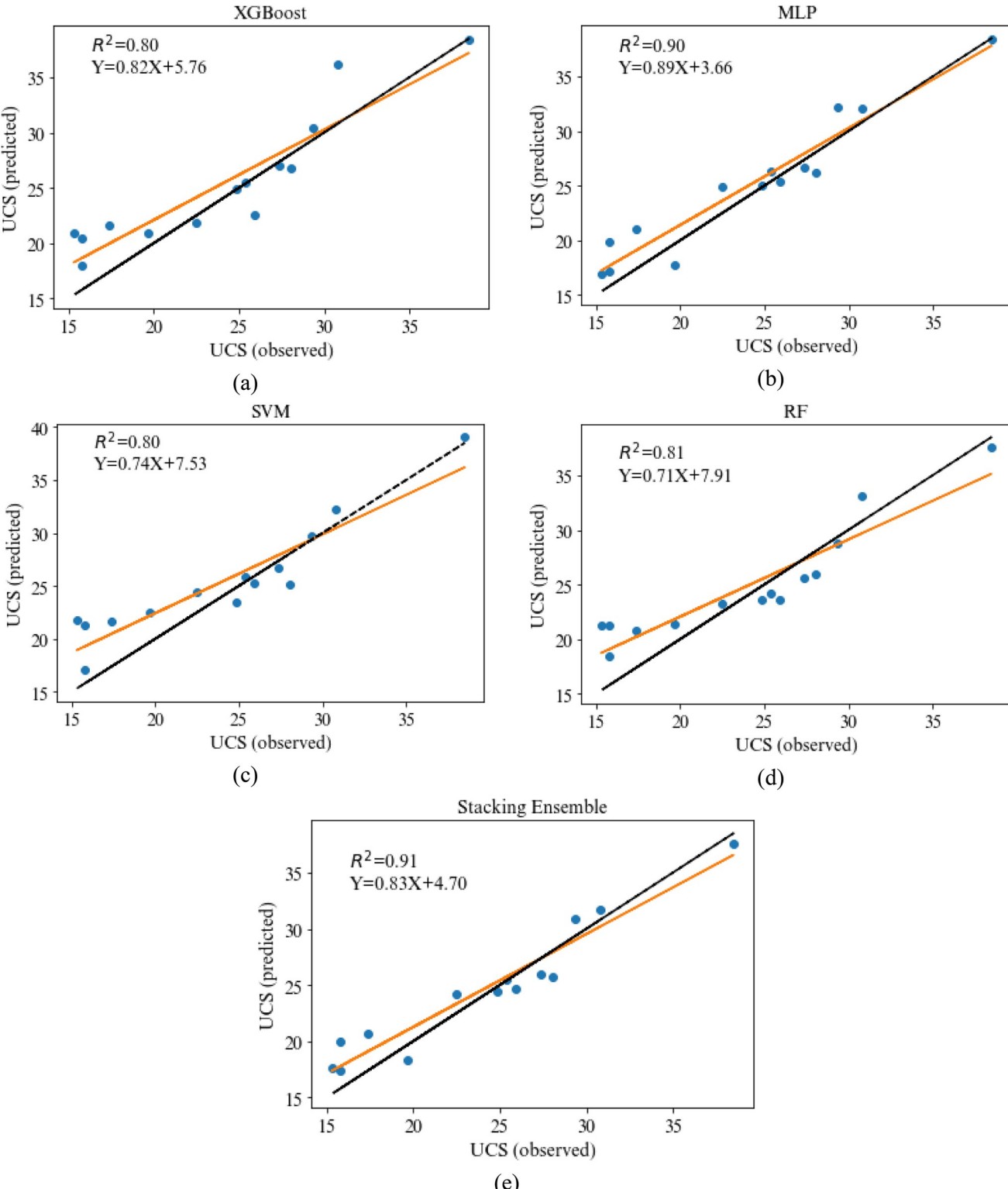

**Fig 16.** Scatter plots of the output of UCS for the a) XGBoost, b) MLP, c) SVR, d) RF, and e) Stacking Ensemble.

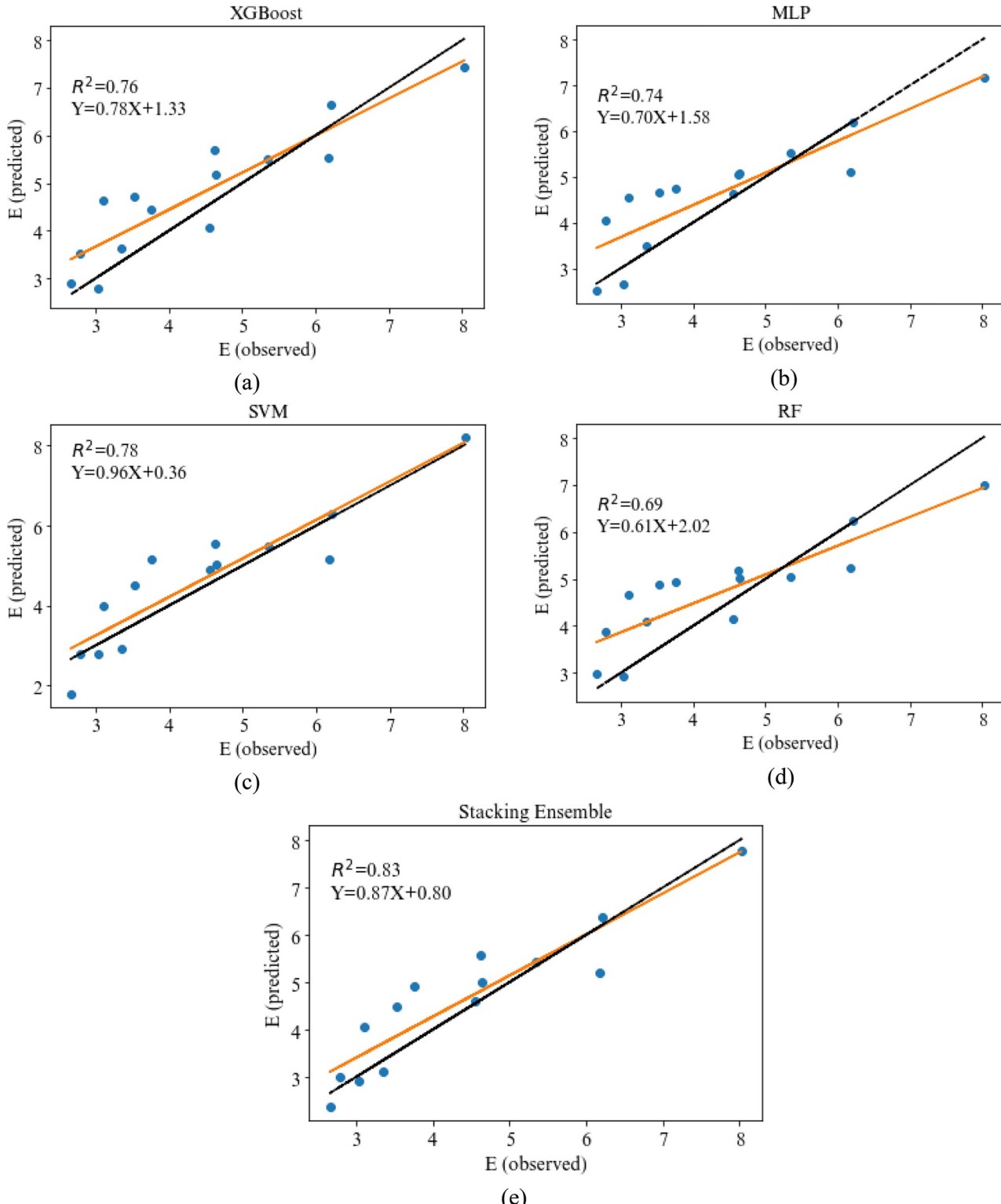

**Fig 17.** Scatter plots of the output of E for the a) XGBoost, b) MLP, c) SVR, d) RF, and e) Stacking Ensemble.

**Table 6. Training time of various models.**

| Method | Training Time of estimating E | Training Time of estimating UCS |
|---|---|---|
| XGBoost | 35.250 | 33.700 |
| MLP | 1.670 | 2.571 |
| SVR | 0.354 | 0.407 |
| RF | 4.890 | 4.985 |
| Stacking Ensemble | 6.798 | 19.514 |

produces predictions that are closer to the true values compared to the base models. However, it is worth noting that the proposed method does not yield satisfactory results in terms of VAF, MAPE, PI, WMAPE, and a20-index when predicting UCS. These metrics indicate that the proposed method explains less variance and has wider prediction intervals compared to other models. Therefore, further research is needed to address this limitation and explore the impact of marginal data on the estimation of UCS.

Additionally, the comparison of training time between the proposed method and other methods supports its suitability from a time perspective. It is important to acknowledge that the proposed method is an ensemble method, which logically results in longer training times compared to single methods such as SVM and MLP. In the comparison with XGBoost, the proposed method demonstrates significantly faster training times. On the other hand, RF exhibits the shortest training time due to its composition of regression trees that can be learned quickly. Furthermore, the parallel learning of the models in the first level can also contribute to reducing the overall training time.

## 5. Conclusions

The most important results and findings of the research can be concluded as follows:

1) The laboratory experiments upon the selected carbonate rocks indicated that they are moderate in strength with low values of the E. They have high surface hardness based on the Brinell hardness test. These rocks have moderate to high $\gamma_d$, very low to moderate $n_e$, and moderate to very high $v_p$ relating to their composing materials and the presence of cavities, veins, and fractures in their textures and structures.

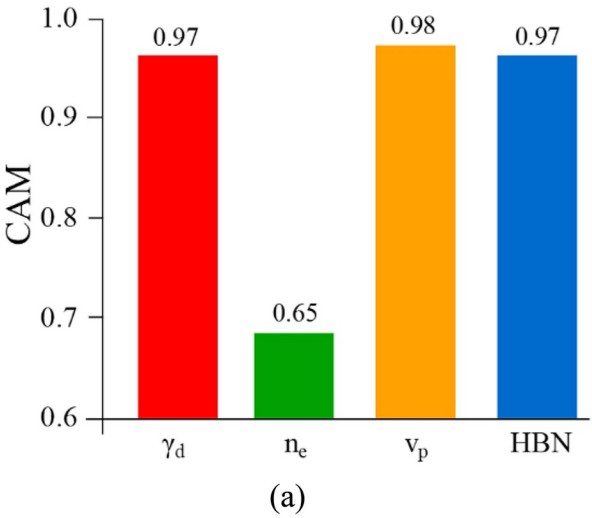
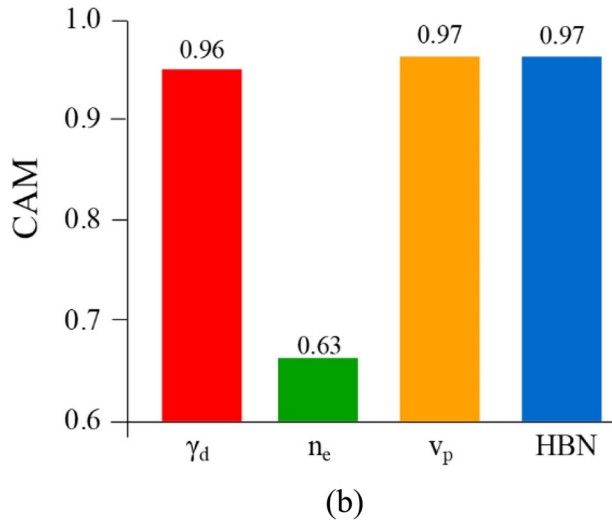

**Fig 18.** RSE of input variables on the a) UCS (MPa), and b) E (GPa).

2) The rocks are recognized very suitable for predicting their engineering properties by using machine learning and soft computing approaches. Therefore, two stacking ensemble methods have been developed for predicting their UCSs and Es from four non-destructive simple laboratory test results namely $\gamma_d$, $n_e$, $v_p$, and HBN, where two base models were considered in the first level of each stacking ensemble.

3) The performance of the developed methods is confirmed in the terms of MAE, MSE, RMSE, and $R^2$. These measures were calculated as 1.657, 3.867, 1.967, and 0.909 for predicting UCS, and 0.483, 0.386, 0.621, and 0.831 for predicting E, respectively. The values of 84.447, 0.130, 0.870, 0.786, 0.115, 1.000, and 0.029 for VAF, MAPE, PI, a20-index, WMAPE, IOA, and IOS further validate the superiority of the proposed method over both base methods, bagging and boosting methods.

4) The obtained results confirm that the exploiting stacking ensemble reduces the error prediction in comparison to the SVR, and MLP as two popular single models. This is due to the fitting ability of individual models can enhance by incorporating them together.

5) Further, our experiments confirm that the stacking ensemble results are superior to the RF and XGBoost as two widely used and successful ensemble models that are based on well-known ensemble methods of bagging and boosting. This is due to the stacking ensemble combines various high-quality models and can correct their errors. 6) It was observed that the stacking ensemble that was suggested did not result in any improvement in the prediction error of UCS on a few performance metrics, which suggests that further research on this topic may be necessary. It is worth to mention, however applying stacking ensemble is more time consuming than using base models alone, but according to the amount of data, the time spent in practice is quite reasonable.

7) Applying suitable stacking ensemble is proposed for predicting other rock properties.

## Acknowledgments

The authors would like to thank Dr. Reza Ahari-Pour for his helps to carry out the microscopic study of thin sections. They also acknowledge the official supports of the Engineering Geology and Rock Mechanics Laboratory of Damghan University for performing all laboratory tests of the research.

## Author Contributions

**Conceptualization:** Davood Fereidooni.

**Formal analysis:** Zohre Karimi, Fatemeh Ghasemi.

**Investigation:** Davood Fereidooni, Fatemeh Ghasemi.

**Methodology:** Davood Fereidooni.

**Project administration:** Davood Fereidooni.

**Software:** Zohre Karimi, Fatemeh Ghasemi.

**Supervision:** Davood Fereidooni.

**Validation:** Zohre Karimi.

**Writing – original draft:** Davood Fereidooni, Zohre Karimi.

**Writing – review & editing:** Davood Fereidooni.

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
