## [Decision Letter · Decision Letter 0]

7 Feb 2024

PONE-D-23-41172Developing two stacking ensemble learning models for predicting uniaxial compressive strength and elasticity modulus of intact carbonate rocks from non-destructive simple laboratory test resultsPLOS ONE

Dear Dr. Fereidooni,

Thank you for submitting your manuscript to PLOS ONE. After careful consideration, we feel that it has merit but does not fully meet PLOS ONE’s publication criteria as it currently stands. Therefore, we invite you to submit a revised version of the manuscript that addresses the points raised during the review process.

**ACADEMIC EDITOR: **The author used machine learning methods to explore rock mechanical parameters. This work has certain engineering significance. Based on the reviewers' comments, I will make a decision on major revisions. I noticed that most of the references in the article are old, and I encourage the author to replace references within the past three years. In addition, the reviewer also gave the corresponding reference index. I suggest that the author can choose whether to cite it based on the article itself. Please add corresponding data and command flow to the paper so that readers can reproduce this work.

We look forward to receiving your revised manuscript.

Kind regards,

Xianggang Cheng

Academic Editor

PLOS ONE

Additional Editor Comments:

The author used machine learning methods to explore rock mechanical parameters. This work has certain engineering significance. Based on the reviewers' comments, I will make a decision on major revisions.

I noticed that most of the references in the article are old, and I encourage the author to replace references within the past three years. In addition, the reviewer also gave the corresponding reference index. I suggest that the author can choose whether to cite it based on the article itself. Please add corresponding data and command flow to the paper so that readers can reproduce this work.

Reviewers' comments:

Reviewer's Responses to Questions

**Comments to the Author**

1. Is the manuscript technically sound, and do the data support the conclusions?

Reviewer #1: Yes

Reviewer #2: Yes

2. Has the statistical analysis been performed appropriately and rigorously? 

Reviewer #1: Yes

Reviewer #2: No

3. Have the authors made all data underlying the findings in their manuscript fully available?

Reviewer #1: Yes

Reviewer #2: Yes

4. Is the manuscript presented in an intelligible fashion and written in standard English?

Reviewer #1: Yes

Reviewer #2: Yes

5. Review Comments to the Author

Reviewer #1: The manuscript provides a prediction method for UCS and E, which is interesting. The writing logic is clear, the content is substantial, and it has some research significance. Before accepting publication, some content can be improved:

1. Each group underwent four experiments to obtain values for "γd and ne." Were these data processed? Were certain experimental data excluded? Are all these data related to predicting rock parameters?

"Four experiments were conducted for each rock, resulting in a total of 280 specimens tested in this step to determine the average values of γd and ne."

2. All variables appearing in equations "(2) and (3)" should be accompanied by units.

3. All variables in the figures should be presented in italics, such as in Figure 5.

4. Before delving into the detailed methodology, please provide an overall technical flowchart for a clearer understanding of the entire rock parameter prediction process.

5. Sections 4.1-4.2 involve data processing; consider whether placing them in the results section would be more appropriate.

6. Text in Figure 9 obscures parts of the image.

7. Some table titles are excessively long; consider simplifying, particularly for Table 3 and Table 4.

8. The discussion section can be added to fully illustrate the limitations, advantages, future research plans, applicability, and improvement suggestions of the proposed prediction method. Some content can be placed in the discussion section, such as:“In spite of high generalization ability and good accuracy of the studied base models ... exploit the advantages of other models.”

9. The manuscript provides a parameter prediction method for a specific type of rock. For other rocks, which parameters may change? I have some interest in this, and some suggestions for model parameter debugging may serve as a good reference for future research.

10. When assessing the superiority of the model, consider not only predictive accuracy but also factors like computation time. The author should provide a comparison of the calculation times or other evaluation parameters for different methods of predicting rock parameters.

11. Consider adding some recent literature related to rock mechanics to enrich the reference material, such as:

[1]10.1007/s10064-017-1210-5

[2]10.1016/j.ijrmms.2017.03.012

[3]10.1016/j.fuel.2023.129584

Reviewer #2: In the present study, the researchers have implemented stacking ensemble, SVR, RF, XGBoost, MLP models to predict the uniaxial compressive strength and elastic modulus of intact carbonated rocks using non-destructive test results. The research is good but revision is needed. My specific comments are as follows:

1. Revise the title to "Non-Destructive Test-Based Assessment of Uniaxial Compressive Strength and Elastic Modulus of Intact Carbonated Rocks using Stacking Ensemble Models."

2. The abstract is well written and presented. It is suggested to include the RMSE results for best model in abstract.

3. The authors are suggested to improve the literature by including published article during 2018-2024. https://doi.org/10.1007/s40515-023-00357-4, https://doi.org/10.1007/s10064-023-03537-1, https://doi.org/10.1007/s41939-023-00191-8, https://doi.org/10.1007/s41939-022-00137-6, etc.

4. Please include the gap found in the literature study, objectives of the present work, and research significance at the end of the introduction section.

5. It is suggested to perform the Pearson’s product moment correlation coefficient, frequency distribution of variables, and descriptive statistics using the following articles: (a) Khatti, J. and Grover, K., 2022. A study of relationship among correlation coefficient, performance, and overfitting using regression analysis. International Journal of Scientific and Engineering Research, 13, pp.1074-1085. (b) https://doi.org/10.22214/ijraset.2022.43662, (c) https://doi.org/10.1007/978-981-19-6774-0_16

6. It is suggested to discuss the research methodology with flow chart in new section 2 Research Methodology. Also, include subsection 4.1 as 3.6 Hyperparameter Tunning (mention the configuration of hyperparameters).

7. The authors used RMSE, MAE, R2, and MSE performance metrics. Any specific reason to implement these metrics. It is suggested to implement VAF, IOS, IOA, a20-index, WMAPE, MAPE, and PI metrics to analyze the results and reliability of models using the following articles: https://doi.org/10.1038/s41598-023-46064-5, https://doi.org/10.1016/j.compgeo.2023.105912, https://doi.org/10.1007/s11831-023-10024-z,

8. It is suggested to revise the sensitivity analysis and give proper citation. Kindly mention “the cosine amplitude method is used to determine the sensitivity of input variables”. Please consider the following articles: https://doi.org/10.12989/cac.2024.33.1.055, https://doi.org/10.1007/s10706-023-02643-x, https://doi.org/10.1016/j.jrmge.2022.12.034

9. Please revise the conclusion section. Report crisp conclusions as per the objectives mapped for the present research. Also, mention the limitations and advantages of developed optimum models.

10. Check the manuscript for axis titles for each graph.

11. Check the complete manuscript for grammatical and punctuation errors.

12. It is strongly suggested to follow the journal guidelines for reference writing and their citation.

The research is technically sound, but a major revision is required before further processing.

6. PLOS authors have the option to publish the peer review history of their article (what does this mean?). If published, this will include your full peer review and any attached files.

Reviewer #1: No

Reviewer #2: No

---

## [Author Response · Author response to Decision Letter 0]

26 Mar 2024

REVISION NOTES

Manuscript number: PONE-D-23-41172

Title: “Developing two stacking ensemble learning models for predicting uniaxial compressive strength and elasticity modulus of intact carbonate rocks from non-destructive simple laboratory test results”

March 17, 2024

Dear Editor,

Thank you and the reviewers very much for your useful and constructing comments and suggestions on the manuscript. After carefully read them, we tried to follow your recommendations as close as possible and have modified the manuscript accordingly. The detailed responses to all of the comments are listed below point by point. The performed corrections are marked by red color in the file of “Manuscript marked-up Version (R1)”. I would be grateful if you would examine the possibility of this paper for publication. I look forward to hear from you as soon as possible. Please do not hesitate to contact me for any more clarifications.

Sincerely yours

Corresponding author

Response to the comments of Editor

Editor Comments:

The author used machine learning methods to explore rock mechanical parameters. This work has certain engineering significance. Based on the reviewers' comments, I will make a decision on major revisions.

I noticed that most of the references in the article are old, and I encourage the author to replace references within the past three years. In addition, the reviewer also gave the corresponding reference index. I suggest that the author can choose whether to cite it based on the article itself. Please add corresponding data and command flow to the paper so that readers can reproduce this work.

Reply; Thank you for your good comment. I would like to inform you that all of the corrections relevant to your comment have been done and marked by red color in the text of the manuscript (please see the file of “Manuscript, marked-up version (R1)”).

Response to the comments of Reviewer #1

Reviewer #1: 

The manuscript provides a prediction method for UCS and E, which is interesting. The writing logic is clear, the content is substantial, and it has some research significance. Before accepting publication, some content can be improved.

Reply; Thank you for your good and detailed comments throughout the manuscript. I would like to inform you that all of the corrections relevant to your comments have been done and marked by red color in the text of the manuscript (please see the file of “Manuscript, marked-up version (R1)”).

1. Each group underwent four experiments to obtain values for "γd and ne." Were these data processed? Were certain experimental data excluded? Are all these data related to predicting rock parameters?

"Four experiments were conducted for each rock, resulting in a total of 280 specimens tested in this step to determine the average values of γd and ne."

Reply; Thank you for your good suggestion. In fact, four experiments were performed for each rock sample and averaging was done from the four obtained numbers. It should be noted that the samples that showed wrong results in the test were repeated again to reduce the test error. The required explain was added to the text.

2. All variables appearing in equations "(2) and (3)" should be accompanied by units.

Reply; Thank you for your precision. The units were added to the text.

3. All variables in the figures should be presented in italics, such as in Figure 5.

Reply; It is a good comment. The mentioned figures were modified.

4. Before delving into the detailed methodology, please provide an overall technical flowchart for a clearer understanding of the entire rock parameter prediction process.

Reply; Thank you for your good and constructing comments. The methodology flowchart of the research was added to the main text as Fig. 1.

5. Sections 4.1-4.2 involve data processing; consider whether placing them in the results section would be more appropriate.

Reply; Thank you. Section 4.1 include the setting of doing experiments and the results are not given in it. Section 4.2 introduce the measures of evaluating the proposed method and don’t include any results. The results are given in sections 4.3 and 4.4. Therefore, while thanking the opinion of the respected referee, the authors prefer the existing segmentation to the proposed segmentation.

6. Text in Figure 9 obscures parts of the image.

Reply; Thank you for your good comments. Figure 9 (Figure 11 in the revised manuscript) was altered in accordance with the hint mentioned by the esteemed referee.

7. Some table titles are excessively long; consider simplifying, particularly for Table 3 and Table 4.

Reply; Thank you for your precision. All tables’ captions have been reviewed to be simple and short enough. The captions of Table 3 and Table 4 (Table 4 and 5 in the revised version) have been modified as the following:

Table 3: Performance results for predicting the UCS (MPa) in the testing phase

Table 4: Performance results for predicting the E (GPa) in the testing phase

8. The discussion section can be added to fully illustrate the limitations, advantages, future research plans, applicability, and improvement suggestions of the proposed prediction method. Some content can be placed in the discussion section, such as: “In spite of high generalization ability and good accuracy of the studied base models ... exploit the advantages of other models.”

Reply; Thank you for your good and constructing comments. The discussion section has been added to the manuscript. The advantages, limitations, applicability, and improvement suggestions have been included in the discussion section. 

9. The manuscript provides a parameter prediction method for a specific type of rock. For other rocks, which parameters may change? I have some interest in this, and some suggestions for model parameter debugging may serve as a good reference for future research.

Reply; Thank you. It is very good comment. In fact, we have tried to use different rock samples to do this research, so that the effect of the used parameters can be better determined. Considering that different rocks have different nature, parameters that are common to all rocks should be used. These parameters can be physical properties of rocks that can be easily measured and can be used to estimate mechanical properties such as uniaxial compressive strength and modulus of elasticity, which are difficult to measure in the laboratory. Based on the findings of this research, in all rock types, the physical properties can be suitable parameters for estimating the mechanical properties. This is previously explained in the introduction part of the manuscript.

10. When assessing the superiority of the model, consider not only predictive accuracy but also factors like computation time. The author should provide a comparison of the calculation times or other evaluation parameters for different methods of predicting rock parameters.

Reply; You are right. The authors have been added more evaluation parameters including training time, VAF, IOS, IOA, a20-index, WMAPE, MAPE, and PI. The results are given in Tables 4, 5, and 6.

11. Consider adding some recent literature related to rock mechanics to enrich the reference material, such as:

[1] 10.1007/s10064-017-1210-5

[2] 10.1016/j.ijrmms.2017.03.012

[3] 10.1016/j.fuel.2023.129584

Reply; Thank you for proposing the suitable references. They were considered and applied on the introduction part to enrich it. 

Response to the comments of Reviewer #2

Reviewer #2: 

In the present study, the researchers have implemented stacking ensemble, SVR, RF, XGBoost, MLP models to predict the uniaxial compressive strength and elastic modulus of intact carbonated rocks using non-destructive test results. The research is good but revision is needed. My specific comments are as follows:

Reply; Thank you for your good and detailed comments throughout the manuscript. I would like to inform you that all of the corrections relevant to your comments have been done and marked by red color in the text of the manuscript (please see the file of “Manuscript, marked-up version (R1)”).

1. Revise the title to "Non-Destructive Test-Based Assessment of Uniaxial Compressive Strength and Elastic Modulus of Intact Carbonated Rocks using Stacking Ensemble Models."

Reply; Thank you for proposing the good title. The manuscript title was revised based on your good suggested title.

2. The abstract is well written and presented. It is suggested to include the RMSE results for best model in abstract.

Reply; Thank you. The abstract has been updated with the inclusion of the RMSE findings.

3. The authors are suggested to improve the literature by including published article during 2018-2024.

https://doi.org/10.1007/s40515-023-00357-4,

https://doi.org/10.1007/s10064-023-03537-1,

https://doi.org/10.1007/s41939-023-00191-8, 

https://doi.org/10.1007/s41939-022-00137-6, etc.

Reply; Thank you for proposing the suitable references. They were considered and applied on the introduction part to enrich it. 

4. Please include the gap found in the literature study, objectives of the present work, and research significance at the end of the introduction section.

Reply; Thank you for your good and constructing comments. The introduction section is modified by the authors in order to provide a clear explanation of the research gap.

5. It is suggested to perform the Pearson’s product moment correlation coefficient, frequency distribution of variables, and descriptive statistics using the following articles: 

(a) Khatti, J. and Grover, K., 2022. A study of relationship among correlation coefficient, performance, and overfitting using regression analysis. International Journal of Scientific and Engineering Research, 13, pp.1074-1085. 

(b) https://doi.org/10.22214/ijraset.2022.43662, 

(c) https://doi.org/10.1007/978-981-19-6774-0_16

Reply; Thanks for your suggestions and the references. Descriptive statistics has been added to the paper in Table 2 and the Pearson’s product moment correlation coefficient is shown in Fig. 3. The frequency distribution of variables is demonstrated in Fig. 2.

6. It is suggested to discuss the research methodology with flow chart in new section 2 Research Methodology. Also, include subsection 4.1 as 3.6 Hyperparameter Tunning (mention the configuration of hyperparameters).

Reply; Thank you for your good and constructing comments. The methodology flowchart of the research was added to section 2 as Fig. 1.

7. The authors used RMSE, MAE, R2, and MSE performance metrics. Any specific reason to implement these metrics. It is suggested to implement VAF, IOS, IOA, a20-index, WMAPE, MAPE, and PI metrics to analyze the results and reliability of models using the following articles: 

https://doi.org/10.1038/s41598-023-46064-5, 

https://doi.org/10.1016/j.compgeo.2023.105912, 

https://doi.org/10.1007/s11831-023-10024-z,

Reply; Thank you for your comment. The method that has been suggested has been assessed using all of the metrics that were mentioned by the reviewer. The outcomes of this evaluation have been presented in Tables 4 and 5. A comprehensive analysis of the results can be found in the discussion section.

8. It is suggested to revise the sensitivity analysis and give proper citation. Kindly mention “the cosine amplitude method is used to determine the sensitivity of input variables”. Please consider the following articles: https://doi.org/10.12989/cac.2024.33.1.055, 

https://doi.org/10.1007/s10706-023-02643-x, 

https://doi.org/10.1016/j.jrmge.2022.12.034

Reply; Thank you for proposing the suitable references. The section on sensitivity analysis has been revised to include the appropriate citation, as instructed by the reviewer (regrettably, the authors were unable to locate the first specified paper).

9. Please revise the conclusion section. Report crisp conclusions as per the objectives mapped for the present research. Also, mention the limitations and advantages of developed optimum models.

Reply; You are right. The results section was revised to cover all the issues considered by the respected referee.

10. Check the manuscript for axis titles for each graph.

Reply; Thank you for proposing the good title. The manuscript title was revised based on your suggested title.

11. Check the complete manuscript for grammatical and punctuation errors.

Reply; Thank you. On the basis of your suggestion, the manuscript has been edited from grammatical and punctuation point of view.

12. It is strongly suggested to follow the journal guidelines for reference writing and their citation.

Reply; Thank you. Required corrections were done.

The research is technically sound, but a major revision is required before further processing.

Finally, the authors would like to thank for your kindly suggestions through the manuscript. 

Best regards

Corresponding Author

---

## [Decision Letter · Decision Letter 1]

10 Apr 2024

PONE-D-23-41172R1Non-destructive test-based assessment of uniaxial compressive strength and elasticity modulus of intact carbonate rocks using stacking ensemble modelsPLOS ONE

Dear Dr. Fereidooni,

Thank you for submitting your manuscript to PLOS ONE. After careful consideration, we feel that it has merit but does not fully meet PLOS ONE’s publication criteria as it currently stands. Therefore, we invite you to submit a revised version of the manuscript that addresses the points raised during the review process.

We look forward to receiving your revised manuscript.

Kind regards,

Xianggang Cheng

Academic Editor

PLOS ONE

Journal Requirements:

Additional Editor Comments:

Hi, Davood Fereidooni, My opinion is to accept after minor revision.

In order to make the reader clear about your conclusion, please list them.

In addition, many of the images in the manuscript are not clear, please replace them with vector graphics.

Please delete at least 6 references from references 38-47.

Reviewers' comments:

Reviewer's Responses to Questions

**Comments to the Author**

1. If the authors have adequately addressed your comments raised in a previous round of review and you feel that this manuscript is now acceptable for publication, you may indicate that here to bypass the “Comments to the Author” section, enter your conflict of interest statement in the “Confidential to Editor” section, and submit your "Accept" recommendation.

Reviewer #1: All comments have been addressed

Reviewer #2: All comments have been addressed

2. Is the manuscript technically sound, and do the data support the conclusions?

Reviewer #1: Yes

Reviewer #2: Yes

3. Has the statistical analysis been performed appropriately and rigorously? 

Reviewer #1: Yes

Reviewer #2: Yes

4. Have the authors made all data underlying the findings in their manuscript fully available?

Reviewer #1: Yes

Reviewer #2: Yes

5. Is the manuscript presented in an intelligible fashion and written in standard English?

Reviewer #1: Yes

Reviewer #2: Yes

6. Review Comments to the Author

**Reviewer #1:** All my concerns have been addressed, and I recommend publishing it. Perhaps this paper will offer some new insights into the uniaxial compression parameters of rocks.

**Reviewer #2: **The authors have incorporated all modifications and corrections well. Therefore, the revised manuscript is accepted.

7. PLOS authors have the option to publish the peer review history of their article (what does this mean?). If published, this will include your full peer review and any attached files.

Reviewer #1: No

Reviewer #2: No

---

## [Author Response · Author response to Decision Letter 1]

11 Apr 2024

Dear Editor,

Thank you and the reviewers very much for your kindness and useful comments and suggestions on the manuscript. After carefully read them, we followed your recommendations and modified the manuscript accordingly. The detailed responses to all of the comments are listed below point by point. The performed corrections are marked by red color in the file of “Manuscript marked-up Version (R2)”. I would be grateful if you would examine the possibility of this paper for publication. I look forward to hear from you as soon as possible. Please do not hesitate to contact me for any more clarifications.

Sincerely yours

Corresponding author

Response to the comments of Editor

Editor Comments:

Journal Requirements:

Reply; Thank you. The manuscript was reviewed for the items mentioned. There are no potential issues that apply to the manuscript.

Additional Editor Comments:

Hi, Davood Fereidooni, my opinion is to accept after minor revision.

Reply; Thank you for your kindness.

In order to make the reader clear about your conclusion, please list them.

Reply; Thank you for your good comment. The conclusion section was presented in the form of numbered paragraphs.

In addition, many of the images in the manuscript are not clear, please replace them with vector graphics.

Reply; Thank you for you good comment. I did not understand which images are not clear. Also, the images cannot be rendered with vector graphics. Anyway, I checked all the images and increased quality of some images if needed.

Please delete at least 6 references from references 38-47.

Reply; Thank you. Seven references were deleted from the list and they removed from the main text of the manuscript. All required correction were done.

Finally, the authors would like to thank for your kindly suggestions through the manuscript. 

Best regards

Corresponding Author

---

## [Editor Report · Decision Letter 2]

15 Apr 2024

Non-destructive test-based assessment of uniaxial compressive strength and elasticity modulus of intact carbonate rocks using stacking ensemble models

PONE-D-23-41172R2

Dear Dr. Fereidooni,

We’re pleased to inform you that your manuscript has been judged scientifically suitable for publication and will be formally accepted for publication once it meets all outstanding technical requirements.

Kind regards,

Xianggang Cheng

Academic Editor

PLOS ONE
---

## [Editor Report · Acceptance letter]

29 Apr 2024

PONE-D-23-41172R2 

PLOS ONE

Dear Dr. Fereidooni, 

I'm pleased to inform you that your manuscript has been deemed suitable for publication in PLOS ONE. Congratulations! Your manuscript is now being handed over to our production team.

Kind regards, 

on behalf of

Dr. Xianggang Cheng 

Academic Editor

PLOS ONE